# Efficient Time Series Processing for Transformers and State-Space Models through Token Merging

## Abstract

Transformer architectures have shown promising results in time series processing. However, despite recent advances in subquadratic attention mechanisms or state-space models, processing very long sequences still imposes significant computational requirements. Token merging, which involves replacing multiple tokens with a single one calculated as their linear combination, has shown to considerably improve the throughput of vision transformer architectures while maintaining accuracy. In this work, we go beyond computer vision and perform the first investigations of token merging in *time series analysis* on both time series transformers and state-space models. We further introduce *local merging*, a domain-specific token merging algorithm that selectively combines tokens within a local neighborhood, achieving two major benefits: a) Local merging can adjust its the computational complexity from quadratic to linear based on the neighborhood size to effectively scale token merging to long sequences; b) Local merging is the first causal merging scheme enabling token merging in transformer decoders. Our comprehensive empirical evaluation demonstrates that token merging offers substantial computational benefits with minimal impact on accuracy across various models and datasets. On the recently proposed Chronos foundation model, we achieve accelerations up to $5400\,\%$ with only minor accuracy degradations.

## 1 Introduction

Since their inception in NLP (Vaswani et al., 2017), transformers have extended their influence into various domains, including computer vision with Vision Transformers (ViTs) (Dosovitskiy et al., 2021), graphs (Yun et al., 2019), and time series processing (Li et al., 2019). However, the computational complexity of the standard attention mechanism used in transformer architectures scales quadratically with the number of input tokens, resulting in high memory requirements. This scalability issue becomes especially pronounced in time series processing, where sequences frequently comprise thousands of tokens (Godahewa et al., 2021). Consequently, recent foundational models in time series, such as Chronos, exhibit impressive zero-shot generalization capabilities but demand substantial computational resources (Ansari et al., 2024).

Recently, state-space models have emerged as a solution to mitigate the computational burden of transformers. Their complexity scales subquadratically with the sequence length (Poli et al., 2023), which allows them to process millions of tokens (Nguyen et al., 2023). However, even in state-space models, very long sequences will impose considerable memory and computational demands.

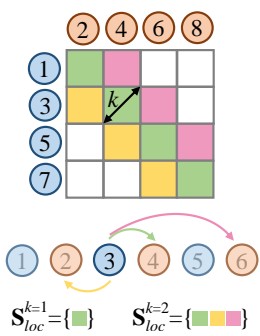

Figure 1: **Local token merging:** Computing token similarity on a subset $\mathbf{S}_{loc}$ under locality constraint $k$ reduces token merging's quadratic complexity to linear.

Bolya et al. (2023), have shown that the efficiency of ViTs can be substantially improved by *merging* tokens throughout the transformer architecture. Specifically, they compute similarity scores between

tokens and combine them into single tokens through a convex combination. However, they only explore token merging for ViT architectures.

In this empirical study, we for the first time explore token merging within the time series domain. We introduce a novel *local* token merging algorithm whose computational complexity varies from quadratic to linear, based on the neighborhood considered for each token merge. This allows token merging to scale to long sequences. Further, *local* merging preserves causality and therefore enables token merging in transformer decoders. The algorithm is illustrated in figure 1. Through comprehensive empirical evaluations, we analyze the impact of token merging on various time series transformer models and state-space models. Our key contributions are as follows:

- **Token merging in time series**  We present first studies on token merging in time series analysis, exploring its application beyond transformer architectures to include state-space models. For this purpose, we propose a domain-specific token merging algorithm that combines tokens within a local neighborhood around each token, preserving causality. Adjusting the size of the neighborhood allows the complexity of the algorithm to range from quadratic to linear. We show that this adaptability improves upon classical token merging for long-sequence modeling tasks. Further, we present first token merging in transformer decoders using our causal merging algorithm.
- **Model acceleration**  Our findings from both pretrained models and those trained with token merging reveal substantial computational savings with only slight reductions in accuracy across five time series transformer architectures and datasets. We also assess five different model sizes per architecture, noting greater relative accelerations in larger models.
- **Token merging in foundation models**  Foundation models are becoming increasingly relevant in time series processing, showing superior zero-shot capabilities compared to existing methods (Garza & Mergenthaler-Canseco, 2023; Das et al., 2023; Rasul et al., 2023; Woo et al., 2024; Ansari et al., 2024). In this context, we enhance the throughput of the time series foundation model Chronos by up to $54.76\times$, with a marginal quality drop of $3\,\%$ in relative MSE using token merging. Across four out of five datasets, we identify Pareto optimal points where token merging simultaneously boosts throughput and increases accuracy. For example, on the ETTh1 dataset, token merging achieves a $14.17\times$ acceleration while the MSE improves by $6\,\%$.
- **Token merging patterns**  Our analysis of token merging's effects identifies three distinct patterns: 1) a consistent decline in performance with an increasing number of merged tokens, 2) initial improvements in accuracy with few merged tokens followed by a drop as merging increases, and 3) scenarios where accuracy remains unchanged regardless of the token merging rate. We investigate all patterns and find different explanations for all behaviors.

## 2  RELATED WORK

**Time series transformers**  In recent years, many transformer architectures with inductive biases for time series have been proposed, successfully outperforming classical and other deep-learning-based methods in time series forecasting quality like recurrent neural networks (Li et al., 2019). Most of them focus on reducing complexity by modifying the attention mechanism. LogTrans uses LogSparse attention (Li et al., 2019), while Informer focuses only on the most relevant queries using ProbSparse attention (Zhou et al., 2021). Additionally, many architectures adopt decomposition techniques to model trend and seasonal patterns (Woo et al., 2022; Wu et al., 2021; Zhou et al., 2022; Liu et al., 2022b). Autoformer leverages autocorrelation as a sequence-based similarity measure in the attention mechanism (Wu et al., 2021). FEDformer uses the frequency domain to model time series effectively (Zhou et al., 2022). Non-stationary Transformers further mitigate the effect of the time series distribution changing over time (Liu et al., 2022b). PatchTST embeds subsequences as tokens to capture local semantic information and reduce complexity (Nie et al., 2023). Other works apply hierarchical attention (Liu et al., 2022a; Cirstea et al., 2022) or leverage attention between the time series variates to better model multivariate patterns (Zhang & Yan, 2023; Liu et al., 2023).

Due to their success in the vision and NLP domain, transformer-based foundation models have recently emerged for time series, often used in zero-shot settings. Many works focus on training transformers directly on large and diverse time series datasets, usually with billions of tokens (Garza & Mergenthaler-Canseco, 2023; Das et al., 2023; Rasul et al., 2023; Woo et al., 2024). Inspired by the success of foundation models in NLP, the recently proposed Chronos model converts continuous time series data into a fixed vocabulary and is trained on both real-world and synthetic data (Ansari et al., 2024). Besides, other research branches focus on fine-tuning vision or NLP models for time

series (Zhou et al., 2023) and on applying large language models directly on time series data (Gruver et al., 2023).

**State-space models**  Due to the quadratic scaling of the attention mechanism, transformer architectures suffer from significant computational cost when processing very long input sequences. Recently, state-space models have shown promising results in overcoming the quadratic complexity of transformers with respect to input length. Linear state-space layers solve the sequential processing requirement of RNNs through linear state-space representations (Gu et al., 2021). The S4 model reduces memory requirements by conditioning the state-space matrix with a low-rank correction (Gu et al., 2022). By using implicit convolutions and a data-aware gating mechanism, Hyena (Poli et al., 2023) became one of the first state-space model architectures to match transformers on NLP tasks. Later work uses hardware-aware algorithms to improve the performance of state-space models on modern accelerators (Gu & Dao, 2023).

**Reducing tokens**  Many works reduce the number of processed tokens to increase the efficiency of transformer architectures in computer vision and NLP, often by pruning (Meng et al., 2022; Goyal et al., 2020). Marin et al. (2021) merge tokens in ViT architectures to reduce the loss of information associated with pruning. Bolya et al. (2023) enhance the token merging algorithm, which they successfully apply to already trained encoder-only models. Besides initial work on classification tasks (Bolya et al., 2023), subsequent work applies token merging to diffusion models (Bolya & Hoffman, 2023). Kim et al. (2024) combine merging and pruning, while other work investigates optimal merging and pruning rates (Bonnaerens & Dambre, 2023; Chen et al., 2023). Concurrent work adapts token merging to preserve the spectral properties of the token space (Tran et al., 2024). However, their merging algorithm still has quadratic complexity, making it unsuitable for long sequence processing.

**Sparse attention and token skipping**  Besides reducing the number of tokens, sparse attention (Child et al., 2019; Li et al., 2019; Zhou et al., 2021; Wu et al., 2021) and token skipping (Raposo et al., 2024) also decrease the computational requirements of transformer models. Sparse attention computes a subset of the attention matrix. Therefore, it can only accelerate the attention mechanism itself and not the subsequent MLP, in contrast to reducing the number of tokens during token merging. According to Marin et al. (2021), this MLP can take over $60\%$ of the total computation in a ViT layer. Further, altering the network architecture from full attention to sparse attention requires a retraining of the model. Concurrent work, such as token skipping (Raposo et al., 2024), involves the selection of a subset of tokens to be processed in a transformer layer. However, it has only been shown in NLP when training from scratch. In contrast to sparse attention and token skipping, token merging can accelerate already trained models and does not require any training data or fine-tuning. This is especially important for recent foundation models, which are expensive to train. In our experiments in sections 5.1 and 5.2, token merging successfully accelerates Informer and Autoformer, which already employ sparse attention. We therefore consider token merging as an orthogonal approach.

Here, we propose a token merging algorithm for the time series domain, which extends beyond previous investigations of token merging in ViTs (Bolya et al., 2023; Bolya & Hoffman, 2023). We systematically evaluate the potential to reduce computational effort in time-series-specific transformer architectures and state-space models.

## 3  TOKEN MERGING

Despite recent advances in efficient transformers, processing long input sequences still induces considerable memory requirements and computational effort. To address this, we propose local merging, an efficient token merging algorithm for state-space models and long sequence processing. Finally, we introduce causal merging as a special case of local merging to allow for token merging in decoder architectures.

**(Global) Token merging in computer vision**  Let a neural network $\mathbf{f}(\mathbf{x}) = \boldsymbol{\Phi}_L \circ \boldsymbol{\Phi}_{L-1} \circ \cdots \circ \boldsymbol{\Phi}_1(\mathbf{x})$ consist of $L$ layers denoted as $\boldsymbol{\Phi}_l$, where each layer takes the output of the previous layer as input. We assume that the input $\mathbf{x}_l \in \mathbb{R}^{t_l \times d}$ consists of $t_l$ tokens with dimension $d$. Thereby, the input tokens are generated by a tokenizer $\mathbf{g} : \mathbb{R}^z \to \mathbb{R}^{t \times d}$ out of $z$-dimensional input data $\mathbf{u}$. In the computer vision domain, $\mathbf{u}$ usually takes the form $\mathbf{u} \in \mathbb{R}^{w \times h \times c}$, where $w, h, c$ are the width, height, and channels of the input image, respectively, and $w \cdot h \cdot c = z$.

To improve the computational efficiency of a given model, Bolya et al. (2023) combine the $r$ most similar tokens in each layer, reducing the tokens to be processed in layer $l+1$ to $t_{l+1} = t_l - r$. Therefore, they split the set of all tokens into two disjoint subsets $\mathcal{A}, \mathcal{B}$ in alternation to avoid merging conflicts and allow for a parallelized computation of merging correspondences. Here $\mathcal{A}$ and $\mathcal{B}$ contain $t_l/2$ elements each, denoted as $\mathbf{a}_i$ and $\mathbf{b}_j$ respectively. The authors compute the cosine similarity between **all** tokens in both subsets $\mathbf{S} = (s_{ij})$ and merge the top $r$ most similar correspondences by averaging the tokens accordingly. This results in a **global** token merging algorithm with **quadratic complexity**. Lastly, the authors use a fixed $r$ to enable batch processing without needing to pad individual batch elements to the same shape after token reduction.

**(Local) Token merging for time series**   In this work, we design token merging mechanisms for time series architectures and demonstrate run-time and even performance improvements over various datasets and models. We assume that the input $\mathbf{u}$ consists of $m$ time stamps with $n$ variates.

Previous work on token merging in image processing explored **global** merging schemes, where every token of each subset $\mathcal{A}$ and $\mathcal{B}$ could be merged with each other (Bolya et al., 2023; Bolya & Hoffman, 2023). However, computing the similarity $\mathbf{S} \in \mathbb{R}^{t_l/2 \times t_l/2}$ between both sets of tokens has a complexity of $O(t_l^2/4)$, which is suboptimal for sequential data often consisting of long token sequences (Godahewa et al., 2021; Grešová et al., 2023), and state-space models featuring subquadratic complexity (Poli et al., 2023; Nguyen et al., 2023).

Therefore, we propose **local merging** - a superset of token merging - by introducing $k \in \mathbb{N}, 1 \leqslant k \leqslant t_l/2$ as a locality constraint where we compute the similarity only on a local subset of tokens $\mathbf{S}_{loc} = \{s_{ij} \mid 1 \leqslant i, j \leqslant t_l/2, |i - j| < k\}$. Figure 1 illustrates the proposed merging algorithm. The locality constraint reduces the complexity to $O(t_l/2 + (k-1)(t_l - k))$. Varying the locality, we achieve **linear complexity** by considering only neighboring tokens for merging up to quadratic complexity by considering a global merging pool, possibly exploiting more redundancy. For efficient computation, we refactor $\mathbf{S}_{loc}$ into a rectangular tensor. An upper bound for the resulting speed up can be given by speed up $\leqslant 3\, L\, 4^{L-1} \cdot (4^L - 1)^{-1}$. The acceleration of deeper models is expected to increase as more subsequent layers can profit from already merged tokens. Local merging additionally preserves order and locality as an inductive bias for sequence processing.

Some time series transformers use processing mechanisms that require a minimum number of tokens in the forward pass. To universally enable token merging in these architectures, we further introduce $q$ as the minimum number of remaining tokens. When encountering odd numbers of tokens $t_l$, we exclude the most recent token with the latest positional embedding from merging as we expect it to contain the most relevant information.

We derive the complexity of the token merging procedures in appendix A.1.

Existing merging schemes are not suitable for causal operations, as global token merging over arbitrary ranges breaks causality. To remedy this limitation and enable token merging in transformer decoders, such as for recent decoder-only foundation models (Das et al., 2023) and encoder-decoder architectures (Ansari et al., 2024), we propose a special case of local merging: By restricting the merging neighborhood to only adjacent tokens with $k = 1$, local merging preserves **causality**.

However, many architectures require a fixed number of decoder output tokens or fixed dimensions for linear projection output layers. To maintain a constant output dimensionality while merging tokens to speed up the decoder, we unmerge all tokens in a final step. Coherent to our causal merging operation, we clone a previously merged token into two neighboring identical ones, to unmerge it. Bolya & Hoffman (2023) propose an unmerging algorithm for computer vision. However, they only leverage non-causal global token merging. Moreover, they immediately unmerge after every merge, which makes it unsuitable for long sequence processing, as it is unable to utilize the cumulative effect of reducing tokens.

## 4 EXPERIMENTS

We systematically explore token merging in diverse settings on 5 time series datasets and 5 model architectures in 5 different sizes each. Additionally, we investigate token merging in large foundation models using Chronos in a zero-shot setting (Ansari et al., 2024). Finally, we demonstrate that token merging can be applied to state-space models for long sequence processing by using a novel local merging algorithm featuring subquadratic complexity.

**Datasets**  We use time series forecasting datasets including ETTh1, ETTm1, Weather, Electricity and Traffic for our transformer experiments. For state-space models, we use the long-range Dummy Mouse Enhancers Ensembl dataset. See appendix A.2 for more details.

**Model architectures**  For our main experiments, we use 5 architectures, including Autoformer, FEDformer, Informer, Non-stationary Transformer, and the vanilla Transformer (Vaswani et al., 2017) as reference. For each model, we evaluate token merging for different model sizes with $L \in \{2, 4, 6, 8, 10\}$ encoder layers, which we train doing hyperparameter optimization (see appendix A.2). We use an input length of $m = 192$, following the results of Nie et al. (2023), and a prediction horizon $p = 96$ samples. Longer sequences would generally benefit token merging.
For experiments on the foundation model Chronos, we use the default input length of $m = 512$ and prediction horizon $p = 64$ (Ansari et al., 2024). We compute the median from Chronos probabilistic forecasts and report the MSE.
For our experiments on state-space models, we use HyenaDNA medium, a genomic foundation model (Nguyen et al., 2023) based on the Hyena architecture (Poli et al., 2023). We use a large input length of $m = 16\,000$ nucleotides utilizing Hyenas subquadratic complexity. We chose Hyena over Mamba (Gu & Dao, 2023) to avoid specialized CUDA kernels and hope to make more general statements about the capabilities of token merging.

**Applying token merging**  In our experiments, we generally find it beneficial to allow self-attention to transfer information between tokens before merging them. Therefore, we apply token merging between self-attention and the MLP in all transformer encoders as Bolya et al. (2023). For our main experiments, we also apply our casual local merging with $k = 1$ in the transformer decoders between self-attention and cross-attention and finally unmerge all decoder tokens. In architectures utilizing additional tensors like attention masks or positional biases, we merge them using the same correspondences. Many transformers exhibit quadratic attention, imposing considerable computational cost. As a result, we do not find the token merging algorithm to introduce a substantially additional overhead. Thus, we choose $k = t_l/2$ to profit from a global merging pool for transformer encoders. Therefore, we utilize different merging strategies in transformer encoders and decoders. In state-space models, we merge tokens after the Hyena operator and choose $k = 1$ to not introduce an operation with quadratic complexity into the architecture.

**Reproducibility of measurements**  We report all results on the same Nvidia A6000 GPU. For training, we utilize Nvidia V100 and A100 GPUs (see appendix A.2). We measure the end-to-end inference time of the models using 2 warm-ups and 2 measurement runs per batch. The standard deviation of the execution time is generally $< 2\,\%$ in our experiments. Besides the inference time as practically most relevant quantity, we report FLOPs as a more hardware independent measure using the thop library (Zhu, 2022). We choose the maximum possible batch size and standardize the results.

## 5 RESULTS

We first present our main results for token merging on pretrained models and models trained with token merging. We then explore token merging in transformer foundation models. Subsequently, we ablate different merging patterns, investigate why token merging improves prediction quality, analyze dependencies on input length, explore the redundancy of input tokens and investigate dynamic merging schemes. Finally, we demonstrate first token merging for state-space models.

### 5.1 TOKEN MERGING IN PRETRAINED MODELS

We investigate token merging in both the encoder and decoder on diverse time series transformer models with different inductive biases. All models are trained on the target dataset and token merging is applied only during inference time, as accelerating already trained models is of high practical relevance. We choose token merging hyperparameters as described in appendix A.2, selecting the fastest token merging trial on the validation set that is within an $0.01$ increase in MSE compared to the reference without token merging. If we do not find a trial with token merging satisfying these tight criteria, we report results without token merging, mimicking how token merging might be applied in practice. We perform all selections on the validation set and report all results on the test set.
The vanilla and Non-stationary Transformers have quadratic attention mechanisms, while the remaining architectures feature subquadratic attention complexities of $O(t_l \cdot \log(t_l))$ for Autoformer and

Informer and $O(t_l)$ for FEDformer. Regardless, our local token merging in the encoder together with our casual token merging in the decoder substantially increase the throughput of most models, up to $3.80\times$, often with no change in forecasting quality, as table 1 shows. In some experiments, token merging even improves the MSE. In line with the formal analysis of potential speed up from token merging conducted in section 3, we generally observe higher accelerations for larger models, as more subsequent layers can profit from already merged tokens. Independent of model size, token merging finds Pareto optimal points in 17 of 25 settings and has no negative effect in the remaining cases.

In some cases, we do not find a model with decent forecasting quality satisfying our criteria. Here, token merging during test only has a larger impact on model accuracy, such as for Autoformer on the Traffic dataset. We address this issue when training with token merging in section 5.2.

Table 1: Token merging speeds up (Accel.) various pretrained transformer architectures of different sizes on several multivariate time series datasets. Merging induces minimal change in quality ($\text{MSE}_\Delta$) compared to the reference without token merging (MSE).

| Dataset | Layers $L$ | Transformer | | | Autoformer | | | FEDformer | | | Informer | | | Nonstationary | | |
|---|---|---|---|---|---|---|---|---|---|---|---|---|---|---|---|---|
| | | MSE | Accel. | $\text{MSE}_\Delta$ | MSE | Accel. | $\text{MSE}_\Delta$ | MSE | Accel. | $\text{MSE}_\Delta$ | MSE | Accel. | $\text{MSE}_\Delta$ | MSE | Accel. | $\text{MSE}_\Delta$ |
| ETTh1 | 2 | 0.75 | 1.38× | 0% | 0.42 | 1.00× | 0% | 0.38 | 1.29× | 0% | 0.87 | 1.40× | 0% | 0.55 | 1.36× | 0% |
| | 4 | 0.71 | 1.81× | 0% | 0.40 | 1.39× | 1% | 0.39 | 1.74× | 0% | 0.92 | 1.30× | 1% | 0.47 | 1.82× | 2% |
| | 6 | 0.66 | 2.33× | 0% | 0.44 | 2.12× | 0% | 0.38 | 2.27× | 0% | 0.93 | 2.39× | 0% | 0.46 | 2.39× | 0% |
| | 8 | 0.84 | 2.90× | 0% | 0.41 | 2.68× | −5% | 0.39 | 2.81× | 0% | 1.23 | 2.20× | 9% | 0.48 | 2.93× | 0% |
| | 10 | 0.69 | 3.51× | 0% | 0.39 | 3.14× | 0% | 0.38 | 3.36× | 0% | 1.16 | 2.45× | 4% | 0.57 | 3.56× | 0% |
| ETTm1 | 2 | 0.52 | 1.35× | 0% | 0.44 | 1.00× | 0% | 0.36 | 1.00× | 0% | 0.65 | 1.40× | 0% | 0.42 | 1.36× | 0% |
| | 4 | 0.58 | 1.85× | 2% | 0.43 | 1.00× | 0% | 0.37 | 1.76× | 2% | 0.60 | 1.78× | −1% | 0.48 | 1.72× | 0% |
| | 6 | 0.62 | 2.11× | 4% | 0.45 | 1.00× | 0% | 0.38 | 1.00× | 0% | 0.59 | 2.16× | −1% | 0.38 | 2.52× | 0% |
| | 8 | 0.60 | 3.09× | 1% | 0.58 | 2.60× | 0% | 0.33 | 1.00× | 0% | 0.61 | 1.61× | 0% | 0.46 | 2.10× | −2% |
| | 10 | 0.62 | 3.72× | 0% | 0.54 | 1.69× | 0% | 0.36 | 1.00× | 0% | 0.57 | 1.00× | 0% | 0.41 | 3.80× | 0% |
| Weather | 2 | 0.25 | 1.44× | −1% | 0.28 | 1.10× | 0% | 0.27 | 1.37× | −2% | 0.35 | 1.43× | −1% | 0.19 | 1.46× | 1% |
| | 4 | 0.28 | 1.95× | 0% | 0.24 | 1.00× | 0% | 0.26 | 1.74× | 0% | 0.24 | 1.89× | 2% | 0.19 | 1.95× | 0% |
| | 6 | 0.28 | 2.19× | 9% | 0.26 | 2.03× | 2% | 0.27 | 2.42× | 0% | 0.21 | 2.19× | 2% | 0.20 | 2.54× | 0% |
| | 8 | 0.32 | 2.20× | 5% | 0.26 | 1.56× | 4% | 0.27 | 2.88× | 0% | 0.30 | 1.56× | 1% | 0.20 | 3.14× | 0% |
| | 10 | 0.35 | 2.49× | 8% | 0.26 | 1.72× | 3% | 0.24 | 1.00× | 0% | 0.31 | 1.69× | 1% | 0.19 | 3.76× | 0% |
| Electricity | 2 | 0.25 | 1.30× | 0% | 0.18 | 1.00× | 0% | 0.20 | 1.24× | 0% | 0.30 | 1.23× | 8% | 0.17 | 1.31× | 0% |
| | 4 | 0.26 | 1.75× | 0% | 0.19 | 1.00× | 0% | 0.19 | 1.64× | 0% | 0.30 | 1.60× | 7% | 0.17 | 1.73× | 1% |
| | 6 | 0.25 | 2.29× | 0% | 0.19 | 1.00× | 0% | 0.20 | 2.22× | 0% | 0.29 | 1.00× | 0% | 0.17 | 2.26× | 0% |
| | 8 | 0.25 | 2.84× | 0% | 0.19 | 1.00× | 0% | 0.20 | 2.72× | 0% | 0.31 | 1.00× | 0% | 0.17 | 2.76× | 0% |
| | 10 | 0.25 | 3.31× | 0% | 0.18 | 1.00× | 0% | 0.20 | 3.33× | 0% | 0.30 | 1.00× | 0% | 0.18 | 2.53× | 7% |
| Traffic | 2 | 0.66 | 1.28× | 1% | 0.63 | 1.00× | 0% | 0.59 | 1.21× | 0% | 0.68 | 1.19× | 6% | 0.60 | 1.27× | 2% |
| | 4 | 0.66 | 1.56× | 3% | 0.60 | 1.00× | 0% | 0.58 | 1.65× | 0% | 0.68 | 1.00× | 0% | 0.59 | 1.68× | 1% |
| | 6 | 0.64 | 2.13× | 1% | 0.61 | 1.00× | 0% | 0.57 | 2.10× | 0% | 0.69 | 1.00× | 0% | 0.62 | 1.58× | 2% |
| | 8 | 0.68 | 2.67× | 0% | 0.60 | 1.00× | 0% | 0.59 | 2.61× | 0% | 0.71 | 1.00× | 0% | 0.59 | 2.69× | 1% |
| | 10 | 0.67 | 3.25× | −1% | 0.59 | 1.00× | 0% | 0.58 | 3.12× | 0% | 0.69 | 1.00× | 0% | 0.59 | 3.16× | 0% |

## 5.2 TOKEN MERGING DURING TRAINING

Here, we apply token merging during training to reduce the model's sensitivity to the algorithm at inference time. As shown in figure 2, models trained with token merging often outperform those trained without it, even if token merging is not applied during testing. This approach enables us to accelerate models such as Autoformer on the Traffic dataset without sacrificing accuracy, which was previously not feasible when applying token merging only during inference. Additionally, token merging accelerates the training process itself by up to $2.27\times$ for Autoformer on the Traffic dataset.

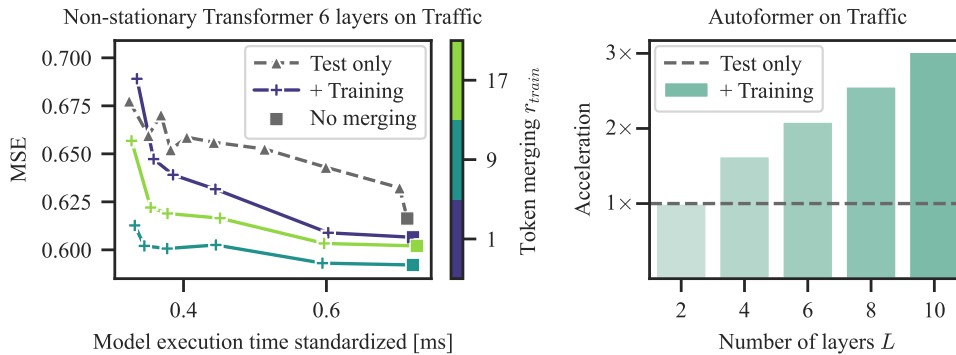

Figure 2: **(Left)** Training with different token merging $r_{train}$ fractions compared to applying token merging only during inference. Even if token merging is not applied during testing (no merging), models trained with token merging achieve better MSE. **(Right)** Additionally, models that showed high MSE degradation with token merging without training show high accelerations while maintaining MSE (increases up to $6\%$) when enabling token merging during training.

Table 2: Token merging acceleration (Accel.) for all Chronos foundation models from tiny to large, measured for zero-shot forecasting of different univariate time series. Applying token merging, we aim for two objectives: the best MSE and the fastest acceleration. Among all Chronos models, we choose the best without token merging as reference (MSE). As token merging improves MSE (negative $MSE_\Delta$) while speeding up the model, we are able to choose small Chronos models while surpassing forecasting quality of larger models.

| Dataset | MSE | Best | | Fastest | |
|---|---|---|---|---|---|
| | | Accel. | $MSE_\Delta$ | Accel. | $MSE_\Delta$ |
| ETTh1 | 0.45 | $14.17\times$ | $-6\%$ | $32.76\times$ | $2\%$ |
| ETTm1 | 0.41 | $1.23\times$ | $-4\%$ | $6.47\times$ | $3\%$ |
| Weather | 0.17 | $1.16\times$ | $-1\%$ | $54.76\times$ | $3\%$ |
| Electricity | 0.14 | $1.02\times$ | $0\%$ | $2.91\times$ | $3\%$ |
| Traffic | 0.61 | $1.16\times$ | $-9\%$ | $2.91\times$ | $1\%$ |

## 5.3 SCALING TO LARGE MODELS

Foundation models are getting more relevant across domains, including NLP (Touvron et al., 2023), computer vision (Kirillov et al., 2023), and time series processing (Das et al., 2023). However, these models have high computational requirements. Therefore, accelerating foundation models without the need for additional fine-tuning is especially important. Thus, we investigate token merging for foundation models on Chronos, a univariate probabilistic model, in zero-shot forecasting setting (Ansari et al., 2024). We apply token merging during inference only, as training Chronos from scratch is not within the scope of this work.

In all our experiments, we find Pareto optimal points with token merging. For four out of five datasets, token merging improves both accuracy and throughput simultaneously (see appendix A.3). Our results demonstrate that it is often beneficial to choose a larger Chronos model with token merging over a smaller one without, as in figure 3. We report our results in table 2, choosing the best Chronos model without token merging as reference. We illustrate two cases: 1) Selecting the token merging setting that provides the best MSE, 2) selecting the setting with the fastest throughput. For 2), we constrain the MSE of token merging trials to be lower than the second-best model without token merging. In addition, we allow a maximum increase in MSE of $3\%$ compared to the reference. In our experiments, we can improve Chronos MSE by up to $9\%$ and speed up inference by $54.76\times$.

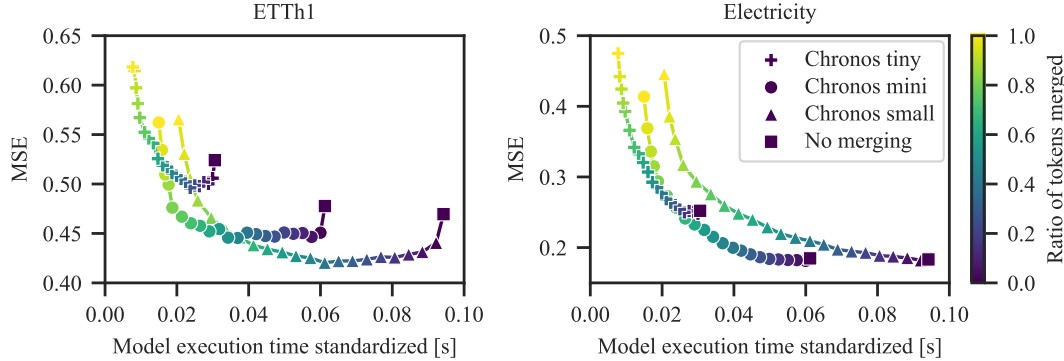

Figure 3: MSE for different token merging in Chronos models during zero-shot testing on two datasets. Choosing larger models with token merging is beneficial compared to smaller ones without.

## 5.4 MERGING PATTERNS

We observe three distinct merging patterns when combining tokens in transformer architectures.

**Increasing MSE**   As the number of merged token increases, the MSE increases almost monotonically (see figure 3). This behavior can be explained due to a loss of information when combining multiple tokens and also occurs in the vision domain (Bolya et al., 2023).

**Constant MSE**   For the vanilla Transformer on ETTh1 and for FEDformer on ETTh1, Weather, Electricity, and Traffic, we observe a constant MSE when applying token merging as shown in figure 4. For the Transformer model, we find all tokens to be similar after the first attention block. Thus, token merging does not affect the model performance. Nevertheless, we find that in most cases, these models still provide reasonable forecasts. In our experiments, transformer models trained on larger or more complex datasets containing more variates do not show this behavior. We argue that this might be a limitation of transformers on small time series datasets (Zeng et al., 2023; Li et al., 2023). Still, token merging successfully improves the throughput while maintaining accuracy for these models.

**Decreasing MSE**   Token merging increases forecasting quality, most prominently in Chronos models as in figure 3. We explain this behavior in section 5.5.

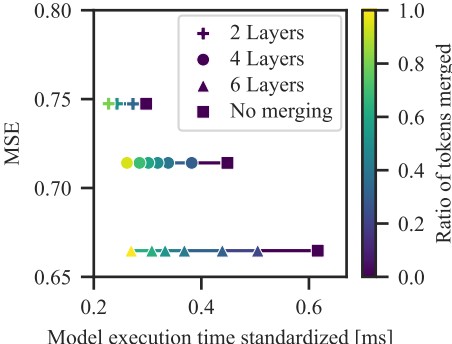

Figure 4: Transformer models on ETTh1 show constant MSE, independent of the amount of token merging $r$.

## 5.5 WHEN DOES TOKEN MERGING IMPROVE MODEL PERFORMANCE

In our experiments, applying token merging sometimes improves MSE. Our hypothesis is that averaging similar tokens smoothes the time series, reducing noise and acting as a low-pass filer. To validate our hypothesis, we low-pass filter the input time series using Gaussian kernels without token merging in figure 5. On ETTh1 and Traffic, both token merging and Gaussian filtering improve the MSE. On the Electricity dataset, token merging and Gaussian filtering do not positively impact the MSE. All of these observations are in line with our hypothesis. Applying token merging together with the Gaussian kernel leads to the best results. Other averaging kernels were significantly worse. We show additional results on ETTm1 and Weather in appendix A.4.

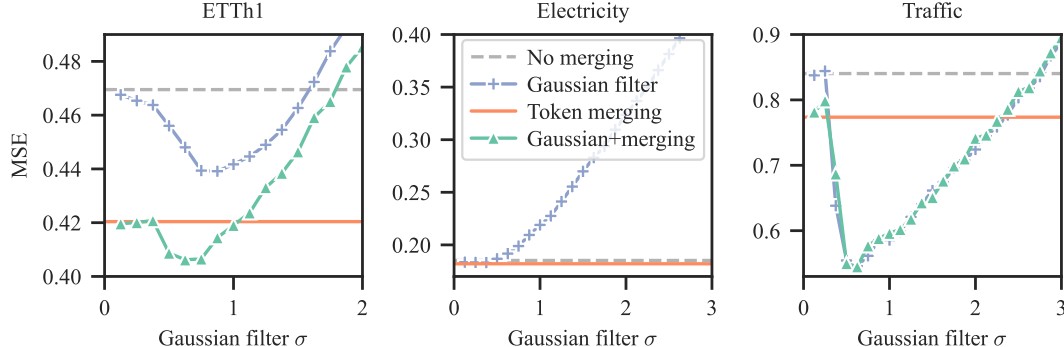

Figure 5: Comparison of the effects of low-pass filtering the input time series with a Gaussian filter and token merging for Chronos small. The Gaussian filter has a similar effect on MSE as token merging, supporting our hypothesis that token merging selectively low-pass filters data. Besides improving MSE, token merging accelerates the model unlike the Gaussian filter.

## 5.6 DEPENDENCIES ON INPUT LENGTH

Token merging effectively reduces the number of tokens in a transformer layer. Here, we explore if we can achieve similar accelerations while maintaining the same prediction quality by varying the number of input samples $m$. For better comparison, we keep the predicted time series snippet fixed and only vary the input sequence.

Our results demonstrate that varying the input length cannot replace token merging (see also appendix A.5). In figure 6, we investigate input length dependence for two objectives in more detail: First, we explore the token merging setup that leads to the best MSE and compare the results to the model without merging. Here, token merging yields considerable throughput increases while improving predictive quality at the same time. Secondly, we compare the fastest model with token merging, which shows no quality decreases, to a standard model. We find models with token merging to scale favorable to long sequences compared to models without merging.

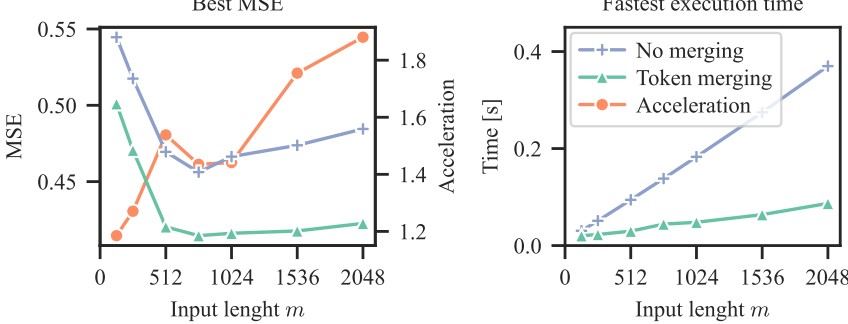

Figure 6: Effect of different input lengths on forecasting quality **(left)** and model execution time **(right)** for token merging in Chronos small models on ETTh1.

## 5.7 REDUNDANCY OF INPUT TOKENS

Token merging exploits similarities in data. Intuitively, the number of tokens that can be merged without affecting predictive performance should depend on the redundancy of the tokens. We explore factors influencing the redundancy of input tokens, including their number and positional embeddings. In the following, we use Autoformer's time stamp positional embedding for our ablation.

First, we investigate whether scaling the number of input tokens increases average redundancy on the ETTh1 dataset. As demonstrated in figure 7a, the same relative number of tokens are merged for a given merging threshold, independent of input length. Therefore, we suggest scaling the number of merged tokens in each layer $r$ linearly with the input length. Positional embeddings add information about the location of a token within a sequence. As a result, two identical tokens without positional

embeddings may show considerable differences when positional embeddings are included, potentially preventing merging. However, figure 7a shows that this effect on token merging is only marginal. It is worth noting that the attention of the transformer acts as a high dimensional low-pass filter, effectively generating more redundancy throughout the transformer layers, as Marin et al. (2021) show. Therefore, token merging not only relies on redundancy of the input data but exploits redundancy that is generated by the transformer itself.

## 5.8 DYNAMIC MERGING

A fixed merging objective allows for batch processing without needing to pad individual time series to the same length. However, it enforces a fixed $r$ among independent batch elements, which might not always be optimal. Determining the number of tokens to be merged dynamically using a similarity threshold might increase quality as no dissimilar tokens are combined. Here, we leverage the single-sample case to explore dynamic merging in optimal conditions. From a practical perspective, this case might be relevant for on-device applications like smartphones or automated driving.

In figure 7b, we compare token merging utilizing a fixed $r$ to dynamic merging varying the cosine similarity threshold. Dynamic merging improves quality slightly in most settings. Therefore, we suggest using a fixed merging schedule for batch applications and dynamic merging just for the single-sample case. There is no equivalent $r$ to dynamic merging schedules as they are similarity-based and strongly layer-dependent. We report FLOPs as we observe substantial execution overhead in time measurements.

## 5.9 TOKEN MERGING IN STATE-SPACE MODELS

State-space models can process very long sequences with millions of tokens due to their subquadratic complexity. Our proposed local merging algorithm is specifically designed to match this subquadratic complexity, enabling effective token merging in state-space models. Additionally, it preserves locality and order as inductive bias for sequence processing.

We compare local and global token merging in HyenaDNA (Grešová et al., 2023), for two objectives: the largest speed up and the best prediction quality. We use a classification task, where the data consists of long genomic sequences with 16 000 nucleotides each. Our local merging with $k = 1$ featuring linear complexity and locality bias outperforms global merging with $k = t_l/2$ and quadratic complexity. Table 3 illustrates that

Table 3: Comparison of **global** and **local** token merging for HyenaDNA on the long sequence Dummy Mouse Enhancers Ensembl dataset. **Best**, second.

| Token merging | Accel. | Accuracy |
|---|---|---|
| No merging | $1.00\times$ | $78.9\,\%$ |
| Local merging$^{\text{fastest}}$ | $\mathbf{3.62\times}$ | $74.0\,\%$ |
| Local merging$^{\text{best}}$ | $1.68\times$ | $\mathbf{80.6\,\%}$ |
| Global merging$^{\text{fastest}}$ | $2.93\times$ | $69.4\,\%$ |
| Global merging$^{\text{best}}$ | $1.15\times$ | $80.2\,\%$ |

local merging achieves substantially larger speed up and better accuracy than global merging. This experiment indicates that architecture and domain-specific biases are important when applying token merging. Local merging accelerates HyenaDNA up to $3.62\times$ with a $4.9\,\%$ decrease in accuracy, whereas global merging substantially reduces the accuracy by $9.5\,\%$. Utilizing less aggressive merging schemes, local merging even boosts accuracy

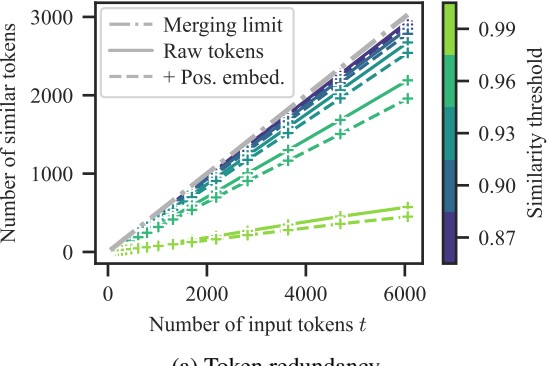

(a) Token redundancy

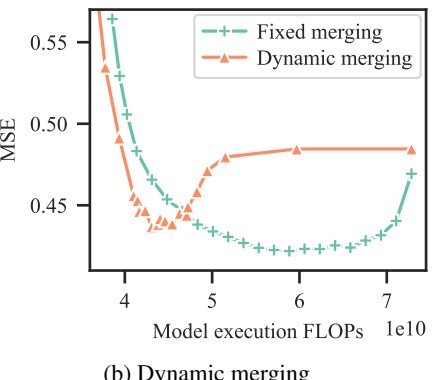

(b) Dynamic merging

Figure 7: **(a)** Relative number of redundant tokens for different similarity thresholds on ETTh1 with and without added positional embedding. **(b)** Comparison of dynamic merging based on a similarity threshold with fixed $r$ merging in single-sample settings for Chronos small on ETTh1.

by $1.7\%$ while still accelerating HyenaDNA $1.68\times$. To the best of our knowledge, this is the first study that investigates merging individual states in state-space models to improve their sequence modeling performance.

# 6 CONCLUSION

In this work, we explore token merging in the time series domain for the first time. We conduct an extensive empirical study on transformer architectures and state-space models in diverse settings using various models and datasets. We demonstrate that token merging can successfully accelerate pretrained models and sometimes even improve their prediction quality. We further introduce a domain-specific *local merging* algorithm with variable complexity and illustrate its effectiveness on the Hyena model. On the long-range Dummy Mouse Enhancers Ensembl dataset, this method outperforms traditional token merging approaches in throughput and accuracy. Additionally, local merging is the first causal token merging scheme, which we successfully demonstrate in transformer decoders. Finally, we conduct several ablation studies to investigate when token merging is most effective, including sequence length, positional embedding, and single-sample inference settings.
We hope that token merging will have a positive effect on reducing the resource consumption and environmental impact of time series models.

**Limitations**  In our work, we divide all tokens into two sets and restrict merging to occur only between tokens from different sets. Future work can explore more flexible merging schemes for time series-specific architectures. Moreover, we do not conduct ablations on all possible hyperparameters due to the large number of architectures and datasets evaluated in this work. Additionally, future work might prioritize past merges or extend locality of merging to periods of the time series. However, the latter leads to non-causal merging. Besides for the time series domain, locality and the linear merging complexity might especially be relevant for high resolution images or videos, which future work can investigate.

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

# A APPENDIX

Supplementary material such as derivations, further details and additional results are listed below.

## A.1 DERIVATIONS

In the following, we derive our theoretical results in section 3.

**Complexity of local merging** To compute $\mathbf{S}_{loc}$ for local merging we need to compute the main diagonal of $\mathbf{S} \in \mathbb{R}^{t_l/2 \times t_l/2}$ and depending on $k$ also secondary diagonals which are symmetrical but shorter than the main diagonal for $k > 1$. We derive the complexity of local merging depending on $k$ in the following:

$$
\begin{aligned}
\text{complexity } \mathbf{S}_{loc} &= \frac{t_l}{2} + 2\sum_{p=2}^{k} \frac{t_l}{2} - (p-1) \\
&= \frac{t_l}{2} + 2\sum_{p=1}^{k-1} \frac{t_l}{2} - p \\
&= \frac{t_l}{2} + 2\left(\frac{(k-1)\,t_l}{2} - \sum_{p=1}^{k-1} p\right) \\
&= \frac{t_l}{2} + 2\left(\frac{(k-1)\,t_l}{2} - (k-1)\frac{k}{2}\right) \\
&= \frac{t_l}{2} + (k-1)(t_l - k)
\end{aligned}
$$

**Merging speed up bound** We roughly estimate the upper bound of the speed up we can achieve by merging tokens in a $L$-layer transformer model. Therefore, we only consider attention due to its quadratic scaling with $t_l$. We disregard additional effects reducing speed up such as merging overhead to estimate the upper bound. Further, we assume merging half of the tokens in each layer. The attention in the first layer is unaffected by merging, as we apply token merging between the attention and MLP.

$$
\begin{aligned}
\text{speed up} &\leqslant \frac{L\,t^2}{t^2 + \left(\frac{t}{2}\right)^2 + \left(\frac{t}{4}\right)^2 + \cdots + \left(\frac{t}{2^{L-2}}\right)^2 + \left(\frac{t}{2^{L-1}}\right)^2} \\
&= \frac{L}{\sum_{p=0}^{L-1}\left(\frac{1}{2^p}\right)^2} \\
&= \frac{L}{\sum_{p=0}^{L-1}\left(\frac{1}{4}\right)^p} \quad \text{using geometric series } \sum_{s=0}^{S} v^s = \frac{1 - v^{S+1}}{1 - v} \text{ for } v \neq 1 \\
&\Rightarrow \frac{L\left(1 - \frac{1}{4}\right)}{1 - \left(\frac{1}{4}\right)^L} \\
&= 3\,L\,4^{L-1} \cdot \left(4^L - 1\right)^{-1}
\end{aligned}
$$

## A.2 Experiments

Here we list additional information concerning our experimental settings and resources.

**Datasets**  We base our experiments on 5 commonly used multivariate time series datasets covering different forecasting applications: *ETTh1* and *ETTm1* consist of 7 variates measuring the power load and temperature of electric transformers in hourly and quarter-hourly granularity (Zhou et al., 2021). *Weather* consists of 21 meteorological quantities such as air temperature and is recorded every 10 minutes in 2020.[1] *Electricity* measures the energy demand of 321 consumers every hour (Godahewa et al., 2021). *Traffic* consists of 862 sensors in the San Francisco Bay Area measuring the road occupancy hourly (Godahewa et al., 2021). We use the same data splits for training, validation and test as Wu et al. (2021) for consistency.

Since the Chronos foundation model operates univariately and requires considerable computational resources, we randomly sample the same 7000 time series from the test set for all Chronos evaluations. For the ETTh1 dataset, we do not observe relevant differences when comparing the results to the full test set.

To explore token merging in an additional sequence-based domain and on a second task, we use the *Dummy Mouse Enhancers Ensembl* dataset (Grešová et al., 2023) for classifying genomic data. It contains very long sequences of nucleotides from a mouse.

**Hyperparameter optimization**  For each transformer architecture, model size, and dataset we train 32 models without token merging doing hyperparameter tuning of *learning rate* and *dropout* using HEBO (Cowen-Rivers et al., 2022). Here, we apply token merging during inference-time only. We choose the best model based on its validation MSE. We train 17 models with the found hyperparameters, the minimum possible $q_{train}$, and different uniformly spaced $r_{train}$ until all tokens are merged. We again choose the best model based on the MSE for further evaluation. We do 185 hyperparameter optimization trials of both chosen models, trained with and without token merging, using HEBO to find token merging inference hyperparameters $r_{test}$ and $q_{test}$ on the validation set. Please note that $r$ and $q$ might be different for local merging in the encoder and causal local merging in the decoder. Finally, we evaluate once on the test set to report our results.

**Hyperparameters**  In table 5 we list the most relevant hyperparameters we used for training the transformer models including the vanilla Transformer, Autoformer, FEDformer, Informer and Non-stationary Transformer. For training and testing HyenaDNA (Nguyen et al., 2023) and for testing Chronos (Ansari et al., 2024) we used their default hyperparameters.

**Computational effort**  We estimate the computational effort for reproducing our experiments in table 4. Please note that we base some of our experiments on model checkpoints acquired in previous experiments.

Table 4: Computational effort to reproduce our experiments.

| Experiment | Accelerator | GPU hours |
|---|---|---|
| Token merging in pretrained models | A6000 | 100 |
|  | V100 | 6720 |
| Token merging during training | A6000 | 50 |
|  | V100 | 3840 |
| Scaling to large models | A6000 | 500 |
| Token merging improves model performance | A6000 | 30 |
| Dependencies on input length | A6000 | 80 |
| Redundancy of input tokens | A6000 | 5 |
| Dynamic merging | A6000 | 140 |
| Token merging in state-space models | A6000 | 40 |
|  | A100 | 6 |

---

[1]https://www.bgc-jena.mpg.de/wetter/

Table 5: Hyperparameters for training the transformer models.

| Hyperparameter | Value |
|---|---|
| **Training** | |
| Seed | 2024 |
| Optimizer | Adam (Kingma & Ba, 2015) |
| Learning rate | Search space loguniform$[10^{-6}, 10^{-2}]$ |
| Learning rate decay | Exponential, $\gamma = 0.97$ |
| Dropout | Search space uniform$[0.0, 0.25]$ |
| Batch size | 32 |
| Epochs | 100 |
| Early stopping patience | 7 |
| Loss | MSE |
| **Model** | |
| Input length | $m = 192$ |
| Prediction horizon | $p = 96$ |
| Token dimension | $d = 512$ |
| Encoder layers | $L \in \{2, 4, 6, 8, 10\}$ |
| Decoder layers | 1 |
| Attention heads | 8 |
| MLP hidden dimension | 2048 |
| Activation | GELU |

### A.3 SCALING TO LARGE MODELS

In this section, we show complete results on applying token merging to Chronos, a time series foundation model.

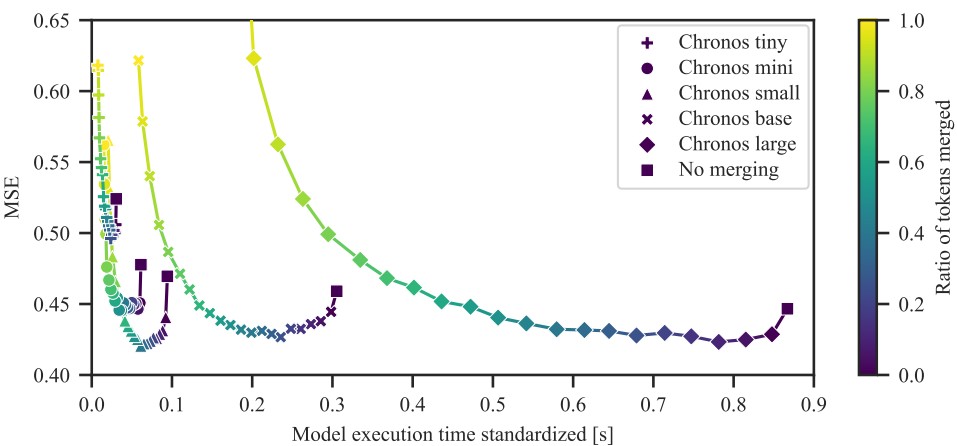

Figure 8: Token merging in different Chronos models on ETTh1

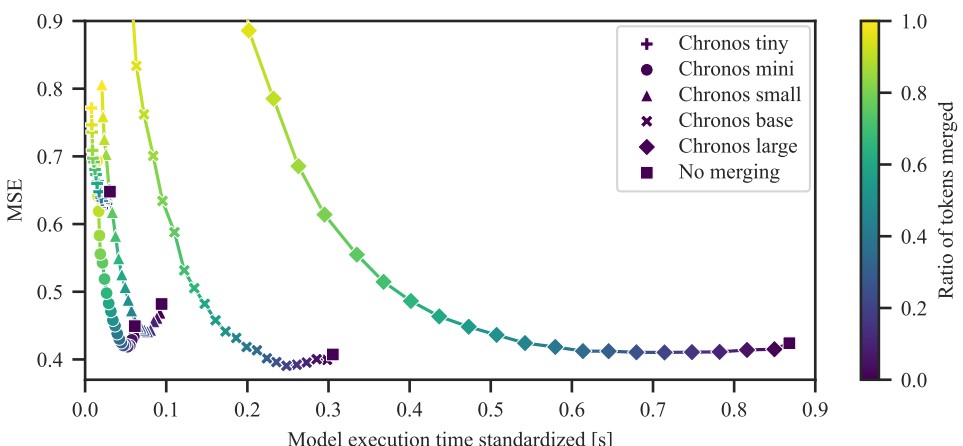

Figure 9: Token merging in different Chronos models on ETTm1

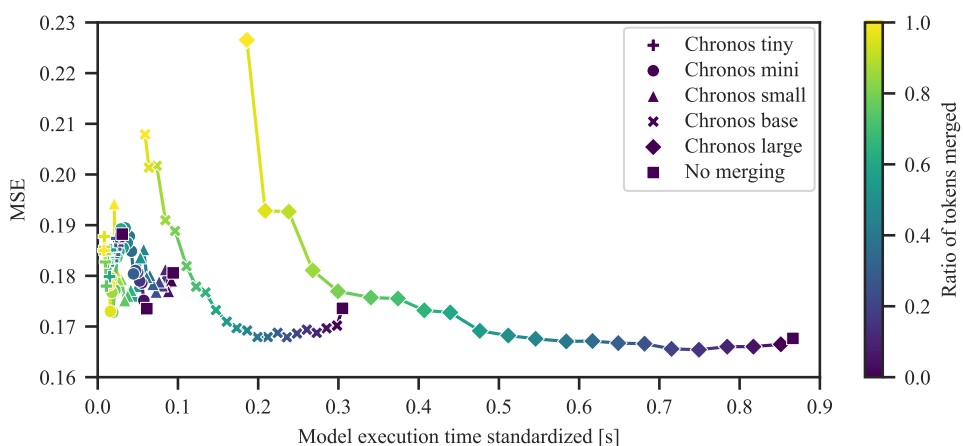

Figure 10: Token merging in different Chronos models on Weather

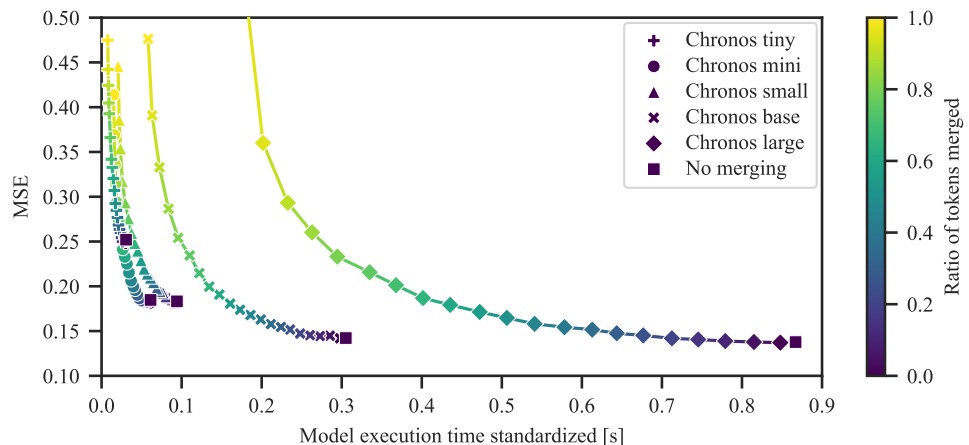

Figure 11: Token merging in different Chronos models on Electricity

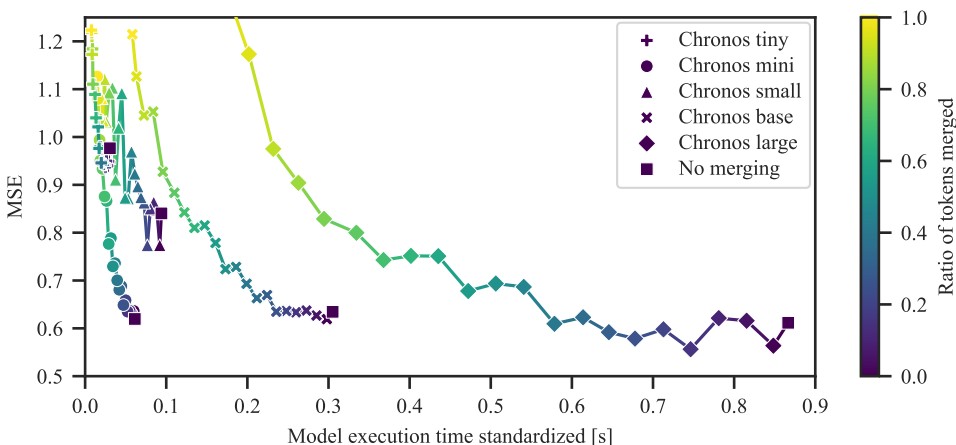

Figure 12: Token merging in different Chronos models on Traffic

## A.4    WHEN DOES TOKEN MERGING IMPROVE MODEL PERFORMANCE

We find token merging to have a smoothing effect improving MSE and show our results on all datasets here.

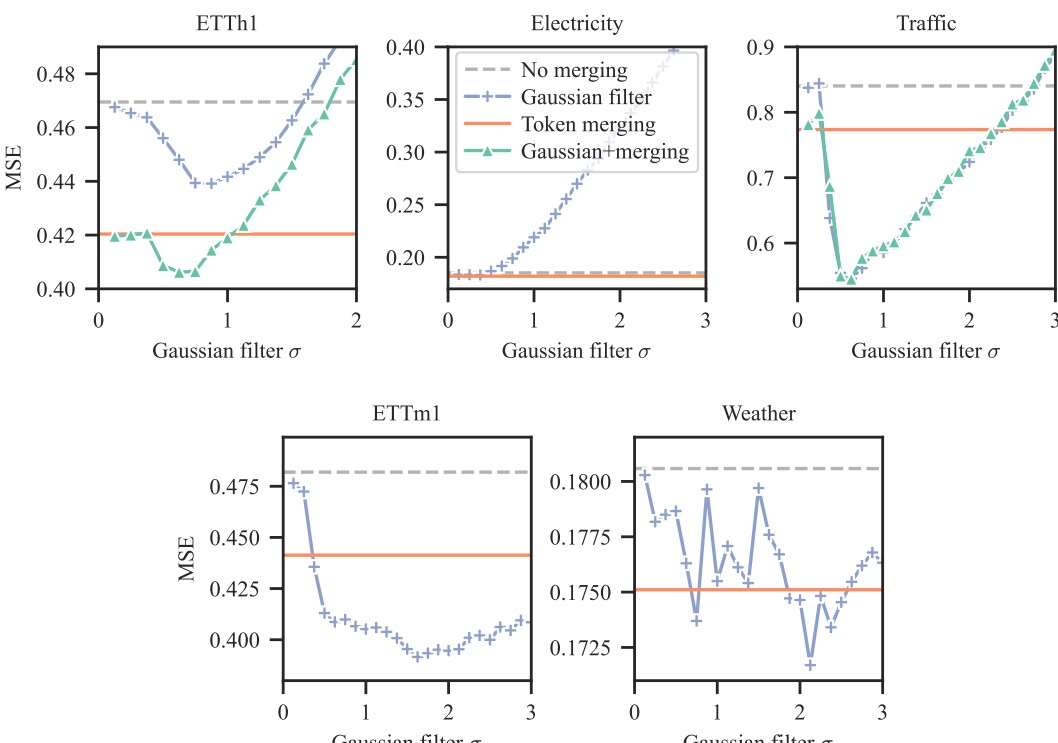

Figure 13: Comparing token merging to smoothing the input time series of Chronos small on different datasets.

A.5    DEPENDENCIES ON INPUT LENGTH

Here we show an additional evaluation on applying token merging in Chronos models with different input lengths.

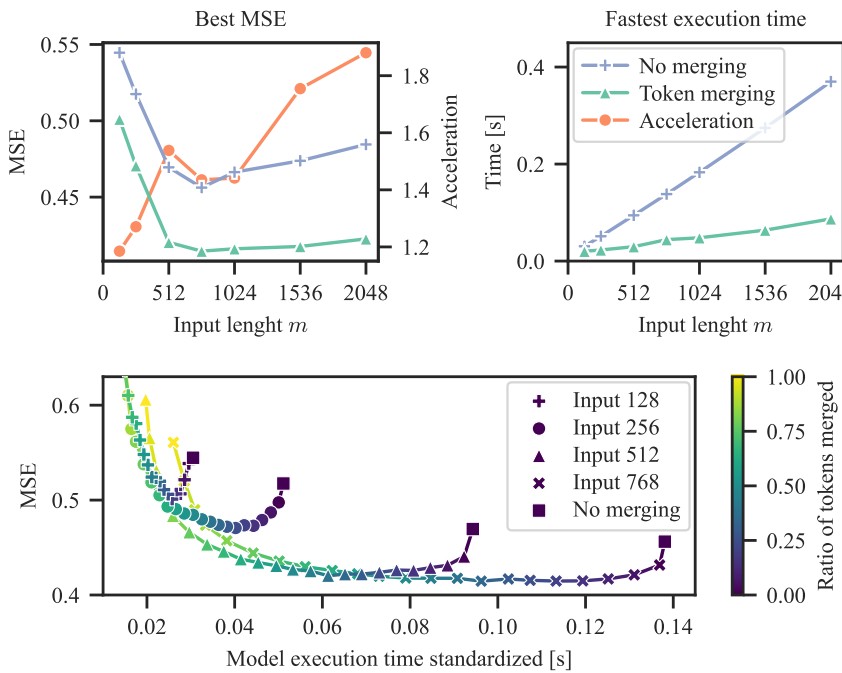

Figure 14: Varying the input length of Chronos small on ETTh1.

### A.6 TOKEN SIMILARITY MEASURES

Different distance measures can be utilized to determine simliar tokens for merging. Here, we explore the $L_1$ and $L_2$ norm as magnitude aware metrics and the cosine similarity measuring the angualar distance. Our results show that the cosine similarity outperforms both, the $L_1$ and $L_2$ norm margnially. Bolya et al. (2023) further ablate the similarity metric for the vision domain.

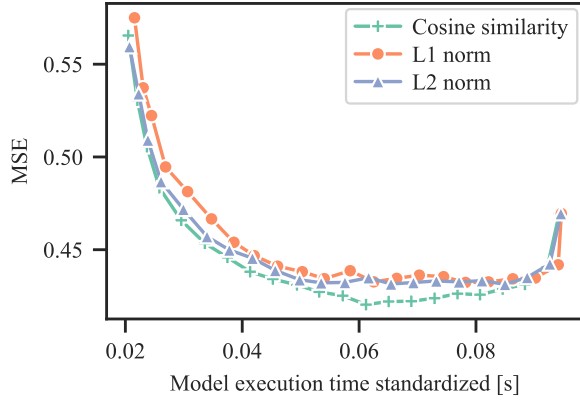

Figure 15: Different token similarity metrics in Chronos small on ETTh1.

## A.7 DATASET PROPERTIES

We find properties of the target dataset that are particularly amenable to token merging. This way we can predict how well token merging will work on a new dataset and gain more insights in the behavior of token merging itself. We find that improvement in forecasting quality due to token merging in table 2 correlate with the spectral entropy of the dataset. Specifically, local merging achieves higher quality gains on high entropy datasets, such as ETTh1, ETTm1 and Traffic (see table 6). We argue that local merging removes unnecessary information from complex signals with high entropy using its selective smoothing ability (see section 5.5). This allows the model to focus on only the relevant patterns of a signal and to achieve better prediction quality. Besides the spectral entropy, the same correlation is evident in the total harmonic distortion. Local merging adaptively low-pass-filters noisy distorted signals to condense the most relevant patterns and effectively improves the signal-to-noise-ratio. The greater noise in ETTh1, ETTm1 and Traffic compared to Weather and Electricity can also be visually inspected in the respective frequency spectrum figures 16 to 20. Therefore, we expect larger improvement of prediction quality when applying local merging on high entropy signals with a low signal-to-noise ratio.

Table 6: Quality improvement due to token merging on datasets with different signal properties.

| Dataset | MSE$_\Delta$ | Spectral entropy | Total harmonic distortion |
|---|---|---|---|
| ETTh1 | $-6\%$ | 4.55 | 54.93 |
| ETTm1 | $-4\%$ | 4.64 | 70.23 |
| Weather | $-1\%$ | 1.64 | 13.15 |
| Electricity | $0\%$ | 2.24 | 15.77 |
| Traffic | $-9\%$ | 2.96 | 19.78 |

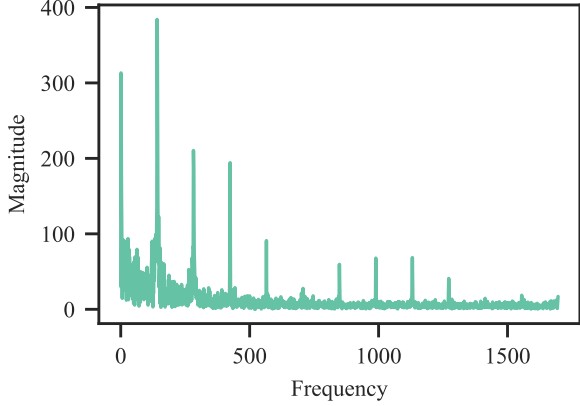

Figure 16: Spectrum of ETTh1.

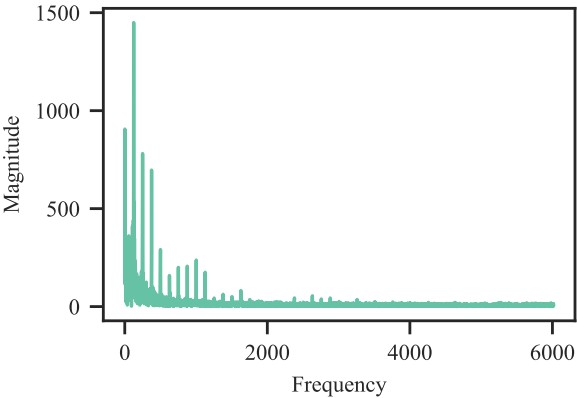

Figure 17: Spectrum of ETTm1.

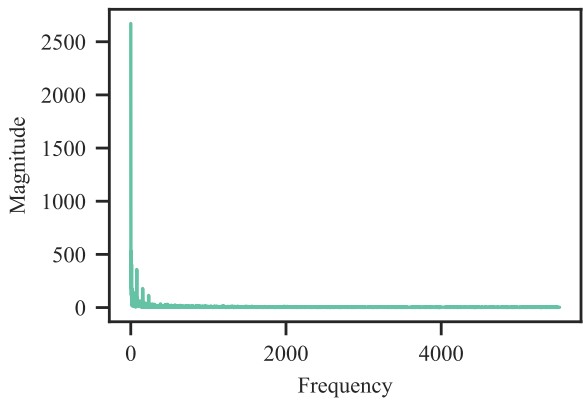

Figure 18: Spectrum of Weather.

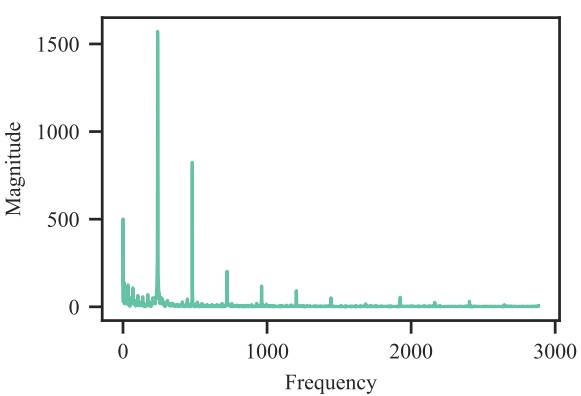

Figure 19: Spectrum of Electricity.

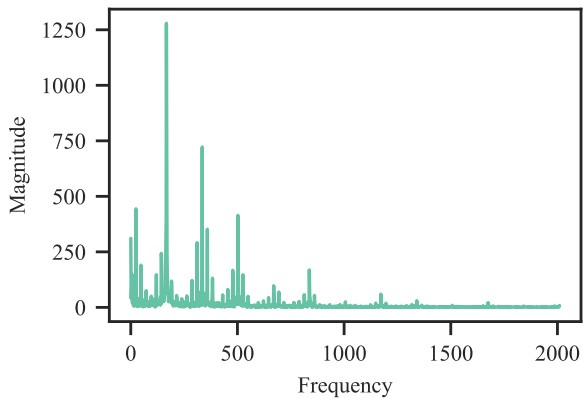

Figure 20: Spectrum of Traffic.

## A.8 MODEL PROPERTIES

Across all datasets, we find properties of the model that are particularly amenable to local merging. For this, we analyze the average cosine similarity of tokens in the models from table 1 after the first transformer layer. We find that local merging accelerates model such as the Nonstationary Transformer, which learn more similar token representations, without quality degradations. Models that show quality degradations when applying local merging like the Informer have learned a dissimilar token representation as table 7 shows.

Table 7: Quality degradation due to token merging of models with different token representations.

| Model and dataset | $MSE_\Delta$ | Token similarity |
|---|---|---|
| Informer 2 Layers Traffic | $6\%$ | 0.10 |
| Informer 4 Layers Electricity | $7\%$ | 0.22 |
| Informer 8 Layers ETTh1 | $9\%$ | 0.28 |
| Informer 6 Layers Weather | $2\%$ | 0.35 |
| Informer 6 Layers ETTm1 | $-1\%$ | 0.40 |
| Nonstationary 10 Layers ETTh1 | $0\%$ | 0.77 |
| Nonstationary 8 Layers ETTh1 | $0\%$ | 0.82 |
| Nonstationary 6 Layers Weather | $0\%$ | 0.87 |
| Transformer 10 Layers ETTm1 | $0\%$ | 0.99 |

