# OpenReview forum: "Efficient Time Series Processing for Transformers and State-Space Models through Token Merging"
_ICLR.cc/2025/Conference — Submitted to ICLR 2025_

### Official Review · Reviewer_P862 · 2024-10-24

**Soundness:** 2
**Presentation:** 3
**Contribution:** 2
**Rating:** 5
**Confidence:** 4

**Summary:**

This paper proposes a token merging technique for time series analysis aimed at improving efficiency. Compared to the global token merging technique (e.g., ToMe), the proposed method reduces merging complexity from quadratic to linear.

**Strengths:**

The paper proposes a token merging scheme for time series analysis and uses extensive experiments to demonstrate the efficiency improvements. Overall, the structure of the paper is clear, and the writing is well-done.

**Weaknesses:**

1.	Originality: The proposed token merging method is straightforward, and similar techniques have been explored in previous works, such as Crossformer (https://openreview.net/forum?id=elZoduqPe8) and CARD (https://openreview.net/forum?id=MJksrOhurE). The claim in the abstract that this paper "performs the first investigations of token merging in time series analysis" seems inaccurate.
2.	Complexity: The linear complexity regarding tokens applies only to the token merging process (which is also the case for existing token merging schemes), rather than to the entire model. This limits the contribution of the paper.
3.	Method: The method section seems too brief and lacks sufficient details, such as how the tokens are merged—are they concatenated or summed?
4.	Experiments:
(1) The naïve token merging method merges k neighboring tokens. Why not compare it with the proposed method?
(2) Large pre-trained models: The proposed method is primarily tested on Chronos. What about other large pre-trained models, such as Timer?
(3) Small models: Regarding efficiency, why not compare with linear-based or CNN-based forecasters (e.g., DLinear)?
(4) Table 1: The base models presented do not seem to include the latest models. What about CARD (ICLR 2024) or Leddam (ICML 2024)?

**Questions:**

Please see weaknesses.

---

> ### Author Response · Authors · 2024-11-20
> **Addressing the Method**
>
> Dear Reviewer P862,
>
> Thank you for taking the time to read our paper and for your valuable comments and suggestions. We are happy to answer
> them in the following:
>
> **Q:** Originality \
> **A:** Crossformer and CARD are both architectures that require training and the presented concepts can not be directly applied to pretrained models. We focus on applying token merging to accelerate already trained models. This is especially important for large foundation models such as Chronos. Therefore, token merging strongly differs from Crossformer, CARD or methods like sparse attention (please see L131-143). Following [1,2,3], we view token merging as an efficiency method which can be applied during inference without requiring any training, training data or fine-tuning. Therefore, we perform first investigations of such a token merging algorithm for time series. Additionally, we propose first token merging with variable complexity for subquadratic models, first token merging for state-space models and first causal token merging for transformer decoders. We thank the reviewer for this comment and will clarify this in the paper.
>
> **Q:** Complexity \
> **A:** You are correct, that the linear complexity is the complexity of the token merging algorithm itself.
> - We focus on applying token merging to accelerate already trained models. To achieve this we can not alter the model architecture itself such as when replacing attention with sparse attention, which would require a retraining of the model (please see L131-143). Therefore, we can not affect the model's complexity itself.
> - Token merging has quadratic complexity by default. We specifically design local merging with linear complexity to accelerate models such as Hyena that  feature subquadratic complexity. Introducing a global merging operation with quadratic complexity to such models would be harmful (please see Sec 5.9).
> - For long sequences, global merging's quadratic complexity introduces a substantial overhead (please see Sec 5.9). Local merging with linear complexity outperforms global merging in both latency and accuracy.
> - Despite the complexity of the model architecture itself, we can accelerate models up to 55x while preserving forecasting quality.
> - Merging tokens throughout the transformer layers has a cumulative effect. This way token merging effectively reduces the complexity of the whole model and enables long sequence processing. We show this in Sec. 5.6 and Sec. 5.9. Models with local merging scale favorable to long sequences.
>
> **Q:** Method: The method section seems too brief
> **A:** We kindly refer to our general comment.
>
>
> **Q:** The naive token merging method merges $k$ neighboring tokens. Why not compare it with the proposed method?
> **A:** We split all tokens into two disjoint sets before computing merging correspondences. This way, we can find all merges without having to iteratively resolve merging conflicts. Merging $k$ neighboring tokens naively would result in merging conflicts. Resolving these conflicts iteratively introduces additional overhead, mitigating the acceleration due to token merging. Therefore, this naive baseline would not be very strong. To challenge local merging in our evaluation, we compare it to sophisticated global merging as a tough baseline and outperform it. We are happy to further discuss your question in case we misinterpreted "naive token merging".

---

> ### Author Response · Authors · 2024-11-20
> **Addressing the Models**
>
> **Q:** Large pre-trained models: What about other large pre-trained models?
> **A:** We chose Chronos as a recent foundation model (from 2024) for our experiments, as it is available in many sizes (from 20M-710M parameters). This way, we demonstrate that token merging scales well even to very large models. Timer only features 29M-67M parameters, which is relatively small in comparison to Chronos. Some of the models in Tab. 1 with 10 Layers have a similar amount of parameters as Timer. Other foundation models such as TimesFM [4] unfortunately did not publish sufficient code to be experimented with. Despite that, we already show 9 experiments on 7 models in 5 sizes and 6 datasets. However, we try to add Timer for the camera ready version.
>
>
> **Q:** Small models: Regarding efficiency, why not compare with linear-based or CNN-based forecasters (e.g., DLinear)?
> **A:** We focus on transformers as almost all current time series foundation models are transformer based [4,5,6,7,8]. Further, transformers are a very universal architecture operating on a set of tokens. This universality allows us to merge tokens to accelerate the model. We evaluate token merging relatively to our baselines without token merging. Despite transformers, we propose first token merging for state space models, which are another promising architecture type. In our experiments we find token merging to be orthogonal to other efficiency methods such as sparse attention.
>
>
> **Q:** Tab. 1: What about CARD (ICLR 2024) or Leddam (ICML 2024)?
> **A:** In Tab. 1 we focused on models, that are well acknowledged in the literature and have been used as baselines in many papers. We hoped to make more comparable statements about the effectiveness of token merging this way. Our baseline models feature different inductive biases for time series, which previous works on vision [1,2] did not take into account. Autoformer and FEDformer use transformations to the frequency domain. Informer and Autoformer already apply sparse attention. We show that token merging can accelerate even these specialized models, which was not evident before.
>
>
> [1] Bolya Daniel, et al. "Token merging: Your vit but faster. In International Conference on Learning Representations", 2023. \
> [2] Bolya Daniel, et al. "Token merging for fast stable diffusion", CVPR Workshop on Efficient Deep Learning for Computer Vision, 2023. \
> [3] Tran, Hoai-Chau, et al. "Accelerating Transformers with Spectrum-Preserving Token Merging." arXiv preprint arXiv:2405.16148 (2024). \
> [4] Das, Abhimanyu, et al. "A decoder-only foundation model for time-series forecasting", arXiv:2310.10688, 2023. \
> [5] Ansari, Abdul Fatir, et al. "Chronos: Learning the language of timeseries", arXiv:2403.07815, 2024. \
> [6] Garza, Azul, et al. "Timegpt-1", arXiv:2310.03589, 2023. \
> [7] Rasul, Kashif, et al. "Lag-llama: Towards foundation models for probabilistic time series forecasting", arXiv:2310.08278, 2023. \
> [8] Woo, et al. "Unified training of universal time series forecasting transformers", arXiv:2402.02592, 2024.

---

> > ### Comment · Reviewer_P862 · 2024-11-26
> >
> > Thank you for your detailed response. I remain positive of this work and will maintain my score.

---

### Official Review · Reviewer_ps2u · 2024-11-02

**Soundness:** 3
**Presentation:** 3
**Contribution:** 1
**Rating:** 3
**Confidence:** 3

**Summary:**

The paper introduces local token merging, a dimensionality-reduction method for very long sequences that can be used to lower the computational cost of training and/or inference in such domains.
Compared to prior work in token merging, the proposed method can be constrained to a local radius (which allows fine-grained control over the computational complexity of the merging process itself), and can be performed in causal manner to support autogressive decoding.
The experiments apply the proposed local token merging to a variety of Transformer and non-Transformer models across a variety of time-series datasets (long sequences but low feature dimensionality), and show that substantial speedup can be achieved at little-to-no detriment to the model accuracy, even when local token merging is only applied during inference.

**Strengths:**

The presentation of the motivation, proposed method, and experiments is clear and easy to understand. The experiments are thorough for the presented scope, evaluating the proposed method across a variety of model architectures/sizes, and datasets/domains. I appreciate the investigation into why local token merging sometimes improves model performance, as this may seem surprising from a first glance.

**Weaknesses:**

The proposed method does not feel like a significant contribution towards the broader goal of more efficient modeling of very long timeseries data. The paper demonstrates the effectiveness of local token merging compared to global token merging, but the experiments/discussion do not effectively place it into the broader context.

I contextualize the proposed method with prior work into two ways:

(1) Methods that can be applied to existing models with no additional training. In this category, I can see the appeal of local token merging, as the experiments show a substantial reduction in computational cost at similar prediction accuracy. However, it seems to me that if applying local token merging is effective in this fashion, then that is an indication that the underlying signal does not contain very much "useful" information (could be either low entropy or low signal-to-noise ratio). This calls into question how broadly applicable this technique can be, if it can only be applied to certain types of data (even within a particular category like "timeseries data"). The comparison between local token merging and gaussian smoothing (Section 5.2; When does token merging improve model performance) exemplifies this point, as local token merging can be viewed as an "adaptive smoothing filter", whereas the gaussian filter requires a fixed window size. Even if it has limited applicability, local token merging could still have be a useful contribution, but then some insight into when/why it is applicable would be helpful to understand its impact.

(2) Methods that are incorporated during training. From this lens, local token merging can be viewed as a particular type of adaptive pooling layer. However, once the model architecture is in scope, then the set of alternatives to local token merging becomes very larger, and the experiments conducted in the paper are not sufficient to evaluate learned token merging in this context. I am particularly curious about more expressive/universal methods like learned tokenizers (continuous VAE for diffusion models [1], discrete autoencoders for multimodal LLMs [2, 3]), which offer significant compression with near-lossless reconstructions. There even exist tokenizers where the degree of compression/dimensionality reduction can be adaptive [4, 5]. One could imagine training a tokenizer (either continuous or discrete) on the timeseries datasets evaluated in this paper, and such a tokenizer could learn a compression algorithm tailored to the timeseries domain, and could employ domain-specific reconstruction metrics analogous to the perceptual losses [6] typically used for images/video. This could potentially outperform local token merging, and the cost of tokenizer training & inference is typically much less than that of the base model.

[1] R. Rombach, et al. High-Resolution Image Synthesis with Latent Diffusion Models. CVPR 2022.

[2] L. Yu, et al. MAGVIT: Masked Generative Video Transformer. CVPR 2023.

[3] D. Kondratyuk, et al. VideoPoet: A Large Language Model for Zero-Shot Video Generation. ICML 2024.

[4] Q. Yu, et al. An Image is Worth 32 Tokens for Reconstruction and Generation. arxiv: 2406.07550.

[5] W. Yan, et al. ElasticTok: Adaptive Tokenization for Image and Video. arXiv:2410.08368.

[6] R. Zhang, et al. The Unreasonable Effectiveness of Deep Features as a Perceptual Metric. CVPR 2018.

**Questions:**

Following the discussion from the the "weaknesses" section:

(1) How can we understand when local token merging will perform well or not? Are there properties of the target dataset/domain that are particularly amenable or hostile to local token merging? Does this change when it is applied only during inference vs also during training?

(2) When training is in scope, how does local token merging compare against other model architectures (e.g. tokenizer + Transformer) that similarly reduce the effective input dimensionality? Depending on the results, what is the broader significance/utility of token merging?

---

> ### Author Response · Authors · 2024-11-20
> **Addressing token merging during inference**
>
> Dear Reviewer ps2u,
>
> Thank you for taking the time to read our paper and for your valuable comments and intriguing ideas. We are happy to answer them in the following:
>
>
> **Q:** Token merging during inference - are there properties of the target dataset/domain that are particularly amenable or hostile to local token merging?  \
> **A:** **We conducted new experiments showcasing properties of the target dataset and properties of the target model that are amenable to local merging.** Please see our general comment. \
> Further, we ablate token merging in many different settings to understand its behavior and when it works best. In most of our experiments, token merging finds Pareto optimal points. Ablations already in our paper:
> - Models: We test 5 transformer models with different inductive biases for time series. Autoformer and FEDformer use transformations to the frequency domain. Informer and Autoformer already apply sparse attention. Token merging can accelerate even these specialized models. Besides that, we show token merging in the foundation model Chronos, which leverages discrete tokens. Further, we present first token merging for state space models.
> - Model sizes: We show local merging on 5 different model sizes ranging from 2 to 24 layers and from ~2M to 710M parameters.
> - Tasks: Local merging performs well on both forecasting and classification (Sec. 5.9)
> - Settings: Local merging can be applied to speed up model training and to make the trained model more robust. However, we focus on applying token merging on already trained models during inference and in zero shot settings.
> - We explain 3 distinctive merging patterns:
>     - Trivial case: Local merging is a tradeoff between acceleration and quality
>     - High cosine similarities inside the transformer lead to constant merging patterns. Here, local merging accelerates the model while preserving quality. (See [1] below: attention generates redundancy throughout the transformer)
>     - Token merging improves quality and accelerates the model at the same time. Here, we show that token merging behaves like a low pass filter.
> - Input length: Models with local merging scale favorable to long sequences. A shorter input length can not replace token merging.
> - Redundancy of input tokens in the dataset:
>     - For different input length and different similarity thresholds, the redundancy scales linearly.
>     - Time position embedding has minimal effect on the redundancy of tokens.
> - Layer-dependent merging strategies: Dynamic merging strategies are superior to a fixed $r$ in single sample settings.
> - Datasets: We evaluate local merging on 5 time series datasets and 1 genomic dataset. Local merging finds Pareto optimal points on all datasets.
>
>
> **Besides, [1] show, that the attention of the transformer acts as a low pass filter, effectively generating more redundancy throughout the transformer layers. Therefore, token merging not only relies on redundancy of the input data but exploits redundancy that is generated natively by the transformer itself, making it more efficient.** We already added this crucial information to our updated paper.
>
> [1] D. Marin, et al. Token pooling in vision transformers. arXiv:2110.03860.

---

> ### Author Response · Authors · 2024-11-20
> **Addressing token merging during training**
>
> **Q:** Token merging during training: other compression methods such as tokenizers  \
> **A:** You are right, that there is a broad spectrum of efficiency methods when training is involved. However, we mainly focus on applying token merging to accelerate already trained models, which is of great importance for emerging foundation models. We experiment with token merging during training only as an ablation to show its potential. Nevertheless, token merging has some advantages compared to other methods such as tokenizer + transformer:
> -  Most importantly, there is no such tokenizer (comparable to NLP etc.) for time series in literature yet. Such a tokenizer would be a different study. We find the idea of tokenizing time series very intriguing.
> - A tokenizer is an extra model, which needs to be fit to the specific dataset. The generalization capability of tokenizers in the time series domain has not been shown. This is of high importance for foundation models during zero-shot inference. We show that token merging generalizes well to unseen data in our Chronos experiments. Further, no extra tokenizer model needs to be trained.
> - A tokenizer requires adjustments to the training pipeline and possibly the model architecture (regression loss -> classification loss). Therefore, a tokenizer is only applicable to some architectures, whereas token merging can accelerate almost all architectures in our experiments. Token merging  requires negligible changes to the training pipeline. As an example:  Autoformer and FEDformer utilize frequency based tokens. A frequency transformation on more complex tokens from a tokenizer is not well defined. Therefore, a tokenizer is not applicable to these architectures whereas token merging accelerates architectures with various inductive biases.
> - Similarly to a tokenizer, token merging accelerates the training process itself by reducing its complexity.
> - We find token merging to perform orthogonal to other efficiency methods like sparse attention (L131-143). We think that token merging might further accelerate a model, which already employs a tokenizer for input compression.
> - In contrast to a tokenizer, which compresses the input, token merging can accelerate every transformer layer with different merging rates (Sec. 5.8). This way it can adaptively reduce complexity throughout the whole transformer and is more flexible than a tokenizer that only works on input data. As an example, for the classification task, only one class token needs to remain after merging in the last layer.
> - Please also see above: Token merging not only relies on redundancy of the input data, but exploits redundancy that the attention mechanism generates.
> - ElasticTok [2] was published in October 2024 after the ICLR paper submission deadline. However, we will add it to our related work section for the camera-ready version.
> - Finally, as we already show 9 experiments on 7 models in 5 sizes and 6 datasets. Therefore, we consider comparing to other vision / NLP methods, that have not yet been shown for time series, out of scope of this work.
>
> [2] W. Yan, et al. ElasticTok: Adaptive Tokenization for Image and Video. arXiv:2410.08368.

---

> > ### Comment · Reviewer_ps2u · 2024-11-26
> >
> > Thank you for your response to my questions and for the additional experiments.
> >
> > However, the results presented in "A.7 Dataset properties" and "A.8 Model properties" do not fully convince me of the significance of local token merging. My interpretation of these results, is that, broadly speaking, (1) local token merging improves MSE when applied to noisy timeseries, and (2) local token merging leads to quality degradations whenever the learned representations vary significantly throughout the feature sequence. These feel like significant restrictions on the applicability of local token merging. Especially since the authors claim that the focus is on accelerating already trained models, it feels like a neat trick rather than a substantial contribution to the field. I would be more convinced of the significance of local token merging by a comparison to more fundamentally different approaches (tokenization, learned modules for upsampling/downsampling, etc).

---

> > > ### Author Response · Authors · 2024-11-26
> > > **Comparison**
> > >
> > > Thanks for the feedback!
> > >
> > > In our empirical evaluation token merging leads to considerable performance improvements for the majority of datasets, indicating that token merging is effective in practice. We argue that the general useability of token merging is **supported** by our new experiments. We observe that token merging is most effective if the data contains some inherent redundancy - a characteristic common in most real-world datasets, which often contain noise or repeated patterns.
> > > Token merging is unlikely to be effective when the input signal lacks any redundancy or noise. However, in such cases, we argue that compressing the signal is theoretically impossible, regardless of the method employed. Here, it is likely that only changes to the model architecture, training algorithm, etc. could improve model performance.
> > >
> > > Yet, one of our main motivations is that token merging can be applied without training, whereas the methods proposed by the reviewer all require additional model training. While this is a minor issue for small models, it makes them unfeasible for most practitioners for large models, such as Chronos, where training is expensive and requires large-scale datasets.
> > >
> > > We appreciate the reviewer's feedback and believe that incorporating a critical discussion on this topic will strengthen the paper.

---

### Official Review · Reviewer_ZtFY · 2024-11-04

**Soundness:** 3
**Presentation:** 3
**Contribution:** 3
**Rating:** 6
**Confidence:** 3

**Summary:**

The authors propose an efficient, customized token-merging strategy tailored for time-series processing in transformer and state-space models. Unlike the "global" token-merging methods used in image domains, the proposed "local" token-merging approach is more efficient and better preserves the causality crucial to time-series tasks. Experimental results demonstrate the efficiency of the proposed method with minimal performance drop across various datasets. Additionally, the method shows the applicability in practical experiments for both transformer-based and state-space model (SSM) approaches.

**Strengths:**

The method is simple, efficient, and effective. Experimental results demonstrate its superior acceleration across datasets and its applicability to both small-scale and large-scale models. Additionally, I appreciate the ablation study, which effectively illustrates the merging patterns and highlights where the proposed method succeeds or falls short.

**Weaknesses:**

My main concerns lie with the clarity of the writing. It is unclear how the authors merge the tokens—are they using averaging, max-pooling, or another method? I believe this should be explained and ablated, especially in relation to temporal issues and handling time-position embedding during token merging. These aspects are not sufficiently clear. Additionally, the authors mention an "unmerging" technique, but it is also unclear how this process is conducted

Furthermore, it is essential to conduct an ablation study on the parameter k to demonstrate the robustness of the proposed method.

**Questions:**

1. See the weakness.
2. I like the dynamic merging results it indeed can improves the FLOPS but I wonder whether it can improve throughputs or latency? The authors should validate it. Besides, it should be able to combine with merging during training setting since the k even can be learned during training.

---

> ### Author Response · Authors · 2024-11-20
> **Addressing the Questions and Weaknesses**
>
> Dear Reviewer ZtFY,
>
> Thank you for taking the time to read our paper and for your valuable comments and suggestions. We are happy to answer them in the following:
>
>
> **Q:** Operation to merge tokens \
> **A:** We kindly refer to our general comment.
>
> **Q:** Operation to unmerge tokens \
> **A:** We kindly refer to our general comment.
>
> **Q:** Temporal issues and handling time-position embedding during token merging \
> **A:** We carefully design local merging with $k=1$ to preserve the casuality of the time series for proper handling of the temporal relationship. In Sec. 5.9 we demonstrate that this locality bias is advantageous over global merging strategies. \
> The additive time-position embedding gets incorporated into the token throughout the transformer layers. The token simultaneously represents its value and position. Therefore, token merging selectively combines tokens based on their value and position (e.g. in periodic series). Further, we ablate in Sec. 5.7 and Fig. 7a whether the temporal position embedding prevents merging and find it has only minor influence.
>
> **Q:** Ablate $k$ \
> **A:** In Sec. 5.9 we compare local merging with $k=1$ to global merging with $k=t_l/2$. Causal local merging outperforms global merging in terms of latency due to its linear complexity. Due to its locality bias preserving causality, local merging is also superior in terms of accuracy. In preliminary experiments we find that either $k=1$ (preserving causality, linear complexity) or $k=t_l/2$ (global merging pool) is best. Other values for $k$ were suboptimal. This is why we choose either $k=1$ or $k=t_l/2$ based on token mergings application.
>
>
> **Q:** Dynamic merging time measurements \
> **A:** We explore the single sample setting in dynamic merging for real-time applications where one can not wait to gather a batch of samples to be processed. As we use batch size 1, the CUDA overhead, which is included in runtime measurements, is strongly GPU dependent. Therefore, we report FLOPs as a more hardware independent measure. However, in preliminary experiments, token merging also improved latency in dynamic merging settings (runtime measurements were just more noisy than FLOPs in this single sample setting).
>
> **Q:** Learning $k$ during training \
> **A:** You are right that $k$ can be learned during training. However, as mentioned above, we find that either $k=t_l/2$ or $k=1$ lead to optimal results in our preliminary experiments. This is why we choose $k$ as a  hyperparameter based on which token merging properties we want to achieve: either causality and linear complexity or utilizing a global merging pool. Further, we focus on applying local merging to already trained models as this is very important for large foundation models, which are expensive to train. We only show token merging during traiing as an ablation to highlight its potential.

---

> > ### Comment · Reviewer_ZtFY · 2024-11-25
> > **Response to the rebuttal**
> >
> > Dear Authors,
> >
> > Thank you for addressing most of my concerns. I recommend merging the detailed illustrations in the revised version to further enhance readability. I have no additional questions and will maintain my score, leaning toward acceptance.
> >
> > Best regards,
> > Chenfeng

---

### Official Review · Reviewer_gi8b · 2024-11-04

**Soundness:** 3
**Presentation:** 4
**Contribution:** 2
**Rating:** 5
**Confidence:** 4

**Summary:**

This work presents a form of token merging to speed up transformers (and state-space models), particularly large and foundational models trained in the time-series domain. It generalises the token merging technique presented by Bolya et al at ICLR 2023 by allowing specifying the number of tokens to be merged (Fig 1, Sec 3), referred to in this work as "local" token merging.

Much of the work is dedicated to studying and analysing the method and presents insightful findings (Figs 2-7; 8-13). The experiments include five architectures on five datasets (many-to-many experiments). The method is shown to speed up inference, which increases with larger models and model architectures, with little to no deterioration in output quality (Tab. 1).

**Strengths:**

1. Firstly, this work addresses a problem relevant to the field in general: attempting to reduce computational complexity of the widespread transform-based and SSM models. Though similar efforts to this end have been made earlier, this work presents some interesting findings, such as choosing larger models with token merging are preferable to smaller ones without (Fig 3) and those trained with token merging scale better to longer observations (Fig 6).
2. The ability of this method to enable its application on models at inference time without having to retrain them with local token merging makes it very applicable. The improved output quality on some smaller models (Tab. 2) is an added strength.
3. This method shows speedup in training while performing comparably to a counterpart trained without token merging (L305). In addition, performance of models so trained without the token merging during inference not sacrificing accuracy is an added incentive to generally consider using this technique.

**Weaknesses:**

1. Obviously, as with the original token merging and its subsequent forms, this method inherits the one same major downside: that of somewhat degraded performance. The more tokens that are merged, the faster the model performs, but the quality of the also tends to decline, and this seems to be a tradeoff (Tab. 2)
2. Another weakness is induced by the nature of the problem: time-series. In order to ensure that tokens are merged while causal relationship between observations is maintaining, setting $k = 1$ in the decoder restricts it from merging a wider set of tokens.
3. Converse to Strength 1, while the relevance of this work is broad, little analysis is done with the representations learnt: namely, the possible use of other measures of similarity measures for merging and unmerging, and the resulting impact on downstream tasks.

    * For instance, if my understanding is correct (see Question 5), it would make more sense to use a quantitatively informed measure to choose tokens to further merge/retain as is, such as measuring the informativeness of tokens or using better heuristics for merging [4]

4. How would this technique translate to video such as human gait analysis or datasets like EPIC Kitchens [3], for example, all time-series? Is there a reason for not experimenting on videos which still offer the causal, time-series essence? Because, such datasets are sufficiently long temporally, offering to provide more insights into Fig 4. (A recent, relevant work [5] might be of interest).
5. Another serious weakness I find is lack of comparison to other token merging algorithms. Have the authors applied current merging algorithms ([1, 2]) to time series just for comparison?

### References

1. Xie, Enze, et al. "SegFormer: Simple and efficient design for semantic segmentation with transformers." Advances in neural information processing systems 34 (2021): 12077-12090.
2. Tran, Hoai-Chau, et al. "Accelerating Transformers with Spectrum-Preserving Token Merging." arXiv preprint arXiv:2405.16148 (2024).
3. Damen, Dima, et al. "Scaling egocentric vision: The epic-kitchens dataset." Proceedings of the European conference on computer vision (ECCV). 2018.
4. Choi, Joonmyung, et al. "vid-TLDR: Training Free Token merging for Light-weight Video Transformer." Proceedings of the IEEE/CVF Conference on Computer Vision and Pattern Recognition. 2024.
5. Lee, Seon Ho, et al. "Video token merging for long-form video understanding." (2024).

**Questions:**

1. Piggybacking on Strength 3, L305 shows speedup in training Autoformer. Is
   this a general trend for all architectures (that have been experimented
   with)?
2. Have other measures of similarity been experimented with (L1, L2, etc)?
3. L021: What is the reason behind suggesting that the local token merging is
   domain-specific? Wouldn't allowing 1 < k < l/2 and relaxing the requirement
   that the tokens must be subsequent make the local ToMe still relevant to
   other domains?
4. Would it be fair to say that Sec 5.5 and Fig 5 support the suggested
   hypothesis provided the datasets (ETTh1, etc) are ridden with considerable
   noise? If this question is invalid, L402/403 that applying token merging plus
   Gaussian leading to the best result is confusing: if on the one hand the
   filter seems to exacerbate the results, and on the other ToMe improves it,
   how come a combination yields the best result?
5. L190: What is meant by "the most recent token"? The token corresponding to
   the latest positional encoding? If that assumption is true, wouldn't it be
   profitable to merge based on a priority, where older tokens are forced to
   merge first?
6. L199: I might have missed this, but if unlike Bolya and Hoffman, how is
   "split a merged token into two neighbouring identical ones" done?

### General comments

* I quite like the separation of the meat of local merging away from the
  experiments. But I found the paragraph from 228-238 somewhat disorganised.
  Starting with "We utilise different merging strategies in transformer encoders
  and decoders" would have make the paragraph clearer to read further. It might
  make a coherent writeup for this paragraph to have generic statements
  (L233-236) in the beginning and specific details afterwards. [This is merely a
  suggestion and not part of any review assessment].

---

> ### Author Response · Authors · 2024-11-20
> **Addressing the Weaknesses**
>
> Dear Reviewer gi8b,
>
> Thank you for taking the time to read our paper and for your valuable comments and questions. We are happy to answer
> them in the following:
>
> **Q:** Weaknesses 1: Token merging is a tradeoff between quality and acceleration \
> **A:** We highlight cases in the Chronos foundation model and in Tab. 1 where token merging simultaneously accelerates the model and boosts quality. For Chronos that is often the case (see Fig. 8-10). Moreover, token merging generally leads to Pareto optimal models, where it can be beneficial to choose a lager model with token merging over a smaller one without. In the majority of our experiments token merging leads to models with faster inference speeds and higher prediction quality. We will clarify this in the paper.
>
> **Q:** Weaknesses 2: Token merging in local neighborhood to preserve causality \
> **A:** We show in Sec. 5.9 that exactly this locality bias is not harmful but beneficial for long sequence processing. Here, causal local merging with $k=1$ outperforms global merging in accuracy. Additionally, local merging with linear complexity is superior to global merging with quadratic complexity in terms of latency. For long sequences, the merging overhead of quadratic global merging is substantial.
> Besides that, we choose the token merging neighborhood size based on its application: for long sequences or in causal decoders, we restrict the neighborhood to also profit from the mentioned locality bias (Sec. 5.9). In transformer encoders, we do not restrict the merging neighborhood and utilize a global merging pool.
>
> **Q:** Weaknesses 3 and Question 2: Other similarity measures / metrics \
> **A:** We kindly refer to our general comment, where we conducted new experiments regarding quantitatively informed token similarity measure.
>
> **Q:** Weaknesses 4: Video datasets \
> **A:** We restrict our analysis to the time series domain, as images and videos are out of the scope of our paper. However, Bolya et al at ICLR 2023 already ablate their global token merging for videos. We see this as a very interesting extension of our work to apply local merging also to other domains. When transferring local merging to the image domain, 2-d image based locality measured would be required.
>
> **Q:** Weaknesses 5: Other token merging approaches \
> **A:** We focus on applying token merging to accelerate already trained models. This is especially important for large foundation models such as Chronos, which are expensive to train. SegFormer [1], however, needs to be trained from scratch and is not applicable to pretrained models. Further, [1,2] are methods designed for ViT encoders and do not preserve causality, which makes them inapplicable to transformer decoders. We specifically design our local merging to preserve the causal properties of time series and present first token merging for decoders.  Additionally, [2] builds a fully connected graph between the tokens to find the ones to merge. This leads to quadratic complexity of their merging algorithm, which the authors list as a limitation. Quadratic complexity merging is unsuitable for long sequence processing or models like Hyena with subquadratic complexity (see Sec. 5.9, where local merging with linear complexity outperforms global merging with quadratic complexity). This is why we design our local merging with varying complexity from quadratic too linear to process long time series and to accelerate models with subquadratic complexity. We already included this discussion in our related work section in our updated paper.

---

> ### Author Response · Authors · 2024-11-20
> **Addressing the Questions**
>
> **Q:** Question 1: Token merging speedup in training Autoformer is this a general trend for all architectures \
> **A:** Yes, you are correct. We experience training speedup when applying token merging during training for all architectures we experiment with in Tab. 1. We also observe less performance degredations (better MSE) from token merging when token merging has already been applied during training.
>
> **Q:** Question 2: We conducted new experiments investigating the token distance measure. Please see above.
>
> **Q:** Question 3: Why we suggest that local merging is domain specific \
> **A:** We suggest, that local merging is domain specific as it is designed to preserve casuality, which is a very important property of time series. But you are right that local merging might be valuable in other domains. Locality and the linear merging complexity might especially be relevant for high resolution images. We included this in our future work of our updated paper.
>
> **Q:** Question 4:
> **A:** In our general comment we conduct new experiments investigating properties of the datasets that are particularly amenable to token merging. You are correct that ETTh1, ETTm1 and Traffic have considerable noise.
>
> **Q:** Question 5: What is meant by "the most recent token"?
> **A:** We kindly refer to our general comment. Thank you for this interesting idea of prioritizing the merges, we already included this in our future work of our updated paper.
>
> **Q:** Question 6: How is "split a merged token into two neighboring identical ones" done?
> **A:** We kindly refer to our general comment.
>
> Thank you also for mentioning paragraph L228-238. We will make our explanation more coherent in the camera-ready version.

---

> ### Comment · Reviewer_gi8b · 2024-11-25
>
> Thanks for answering my questions. For the most part, things are clearer to me and I am satisfied with the answers. Thanks for Fig 15, and thanks for pointing out the experiments in Bolya's work. The reason for pointing out Weakness 5 was to get a broad picture of comparison, but I quite agree that current methods are either for other domains or don't fall within this work's scope to compare with.
>
> As for Weakness 4, the reason I wondered how local merging would do on videos ties to the well-articulated Weakness 1 by Reviewer ps2u, particularly about the breadth of domains to which it can be applied. For my part, Sec. A.7 and the general rebuttal broadly answer this; however, if the gains as pronounced as in the new experiments are to be attributed to the information contained in the target dataset (high entropy) and merging is removing unnecessary information (assuming that information is thanks to a low signal-to-noise ratio), this is hinging upon the fact that there is sufficient noise to remove--a plus for local merging. But then, at the same time, my question is, wouldn't local merging negatively impact by losing supposedly-useless information if the entropy as well as the signal-to-noise ratio were high (a richer and less-noisier target)?

---

> > ### Author Response · Authors · 2024-11-26
> > **Thanks for the feedback**
> >
> > Thank you for your thoughtful feedback and for engaging on a deeper discussion on our work.
> >
> > Regarding Weakness 4, we agree that token merging is unlikely to be effective for datasets with high signal-to-noise ratios and minimal redundancy. Its strength lies in removing redundant or unnecessary information, and applying it to compact, low-noise data could indeed discard valuable information, reducing performance.
> >
> > That said, we argue that most real-world datasets inherently exhibit some level of redundancy or noise due to factors like measurement errors or repetitive patterns. This makes token merging effective in practice across a wide range of datasets, as shown by our empirical results.
> >
> > However, for highly compact datasets, we believe this limitation extends to most test-time efficiency methods, not just token merging.
> >
> > We hope this addresses your concern, and appreciate your effort.

---

> > > ### Comment · Reviewer_gi8b · 2024-11-26
> > >
> > > Thanks. I think that that point must be made clear in the paper, at least for comprehensiveness.

---

### Author Response · Authors · 2024-11-20
**General Comment - Question to the Reviewers**

### Question to the Reviewers
**We included our new ablations and findings in the appendix on pages 22-25. We kindly ask the Reviewers for their opinion on which new experiments to include in the main text of our paper.** Besides our experiments, we already included many of the Reviewers comments in our paper (marked in red). For the camera-ready version we will shorten our paper again to 10 pages.

---

### Author Response · Authors · 2024-11-20
**General Comment - Additional Experiments**

### Additional experiments:

**Other similarity measures / metrics** \
We explore different metrics to determine token similarity during token merging. In our paper we follow Bolya et al at ICLR 2023 using the cosine similarity. In our new experiment, we also include the L1/L2 norm as magnitude aware metrics. We use Chronos small on ETTh1 as base model. Our experiment shows that the cosine similarity outperforms both, the L1 and L2 norm. For further ablation, we kindly refer to the vision token merging paper by Bolya et al at ICLR 2023 (Tab. 1b). Please see page 22 for the results in our updated paper.

**Properties of the target dataset that are particularly amenable or hostile to local token merging** \
In this analysis, we correlate the quality gains due to token merging with properties of the target dataset. This way we can predict how well token merging will work on a new dataset and gain more insights in the behavior of token merging itself. We find that gains in forecasting quality due to token merging (see Tab. 2, Best) correlate with the spectral entropy of the dataset. Specifically, local merging achieves higher quality gains on high entropy datasets, such as ETTh1, ETTm1 and Traffic. We argue that local merging removes unnecessary information from complex signals with high entropy using its selective smoothing ability (Sec. 5.5). This allows the model to focus on only the relevant patterns of a signal and to achieve better prediction quality. Besides the spectral entropy, the same correlation is evident in the total harmonic distortion. Local merging adaptively low-pass-filters noisy distorted signals to condense the most relevant patterns and effectively improves the signal-to-noise-ratio. This greater noise in ETTh1, ETTm1 and Traffic compared to Weather and Electricity can also be visually inspected in the respective frequency spectrum. Therefore, we expect larger prediction quality gains when applying local merging on high entropy signals with a low signal-to-noise ratio. The results are summarized in the following table. Please also see pages 23, 24 in our updated paper.



| Dataset     | MSE$_{\Delta}$ (negative is better) | Spectral Entropy | Total Harmonic Distortion |
| ----------- | ----------------------------------- | ---------------- | ------------------------- |
| ETTh1       | -6%                                 | 4.55             | 54.93                     |
| ETTm1       | -4%                                 | 4.64             | 70.23                     |
| Weather     | -1%                                 | 1.64             | 13.15                     |
| Electricity | 0%                                  | 2.24             | 15.77                     |
| Traffic     | -9%                                 | 2.96             | 19.78                     |

\
**Properties of the target model that are particularly amenable or hostile to local token merging** \
Besides properties of the dataset, we find properties of the model that are amenable to local merging. We analyze the average cosine similarity of tokens in the models (Tab. 1) after the first transformer layer. We find that local merging accelerates model such as the Nonstationary Transformer, which learn more similar token representations, without quality degradations. Models that show quality degradations when applying local merging like the Informer have learned a dissimilar token representation. Therefore, local merging favors models that learn similar token representations.



| Model                          | Quality degradation (see Tab. 1)                          | Token similarity |
| ------------------------------ | -------------------------------------------------------- | ---------------- |
| Informer 2 Layers Traffic      | 6%                                                       | 0.10             |
| Informer 4 Layers  Electricity | 7%                                                       | 0.22             |
| Informer 8 Layers  ETTh1       | 9%                                                       | 0.28             |
| Informer 6 Layers  Weather     | 2%                                                       | 0.35             |
| Informer 6 Layers ETTm1        | -1%                                                      | 0.40             |
| Nonstationary 10 Layers ETTh1  | 0%                                                       | 0.77             |
| Nonstationary 8 Layers  ETTh1  | 0%                                                       | 0.82             |
| Nonstationary 6 Layers Weather | 0%                                                       | 0.87             |
| Transformer 10 Layers ETTm1    | 0%                                                       | 0.99             |

---

### Author Response · Authors · 2024-11-20
**General Comment**

Dear Reviewers,

Thank you for your valuable feedback and effort. We've made several improvements to our work and incorporated them in the updated version of our paper. We are happy to discuss any open questions. In this general comment, we would like to further elaborate the exact operation to merge and to unmerge tokens.

**Q:** Operation to merge tokens \
**A:** You are correct, we merge tokens by averaging them. Due to the computational cost of the performed experiments we had to prioritize, which design decisions are most important to ablate. In this case, we rely on previous results from Bolya et al at ICLR 2023, who ablate stacking / concatenation, averaging and keeping one token / max-pooling in their Tab. 1c,d and find that averaging is most effective. We focus our ablations on time series specific details of token merging.

**Q:** Operation to unmerge tokens \
**A:** Coherent with our merging operation (averaging) of two tokens, we unmerge them. We clone / copy a previously merged token $t$ to two neighboring tokens $t_1 \leftarrow t$ and $t_2 \leftarrow t$. This way we preserve causality. Additionally, we unmerge the tokens after the last transformer layer to utilize the cumulative effect of reducing tokens for long sequence processing. In contrast, Bolya and Hoffman do not preserve causality and unmerge immediately after every merge, which would introduce a large computational overhead in long sequence processing.

**Q:** What is meant by "the most recent token"?
**A:** Thanks for pointing this out. You are correct, we exclude the token with the latest positional encoding / the most recent time series sample / measurement from merging when encountering odd numbers of tokens. It in also an intriguing idea to merge tokens based on a priority, i.e., forcing to merge older tokens first. We think that this also depends on the time series itself. In some cases, past events might be more relevant than more recent data (such as for periodic data).

---

### Author Response · Authors · 2024-11-25
**Friendly Reminder**

We are writing to kindly check if you've had the chance to review our rebuttal response. We believe we could adress most of the raised concerns. If you have any remaining questions or concerns, we'd be happy to address them.

Thank you for your time and consideration.

---

### Meta-Review · Area_Chair_wug7 · 2024-12-22

**Metareview:**

The paper proposes a local token merging method aimed at improving computational efficiency in time-series analysis for transformer and state-space models. The key advantage is that comparing to global merging whose complexity is quadratic, local merging reduces the merge complexity to linear, enabling efficient processing of long time-series data. The merging process respects the causality inherent in time-series tasks, making it applicable for autoregressive decoding. The method can also be applied during inference without retraining the model. On the negative side, the reviewers are concerned how broadly applicable this technique can be and whether it can only be applied to certain types of data. The reviewers appreciate that local token merging improves MSE when applied to noisy timeseries, but they are worried that the quality will degrade whenever the learned representations vary significantly throughout the feature sequence, both of which are hard to tell before testing. There has been extensive message exchange among the expert reviewers and the authors. However, after thorough discussion, the reviewers still find the cons outweigh the pros. The ACs agree. The authors are encouraged to take into account the feedbacks and re-submit to a future venue.

**Additional Comments On Reviewer Discussion:**

The reviewers raised several questions, including the details of operations, the applicability of the approach, the ablation of the design, etc. The authors managed to address most of them during the discussion period. However, while many concerns are gone, the reviewers are still worried about the significance of local merging and intend to keep their original (negative) score.

---

### Decision · Program_Chairs · 2025-01-22

Reject